# China's annual forest age dataset at 30 m spatial resolution from 1986 to 2022

Rong Shang[1, 2], Xudong Lin[1, *], Jing M. Chen[1, 3, *], Yunjian Liang[1], Keyan Fang[1], Mingzhu Xu[1], Yulin Yan[1], Weimin Ju[4], Guirui Yu[5], Nianpeng He[5], Li Xu[5], Liangyun Liu[6], Jing Li[7], Wang Li[7], Jun Zhai[8], Zhongmin Hu[9]

[1]Key Laboratory of Humid Subtropical Eco-Geographical Process of Ministry of Education, School of Geographical Sciences, Fujian Normal University, Fuzhou, 350117, China
[2]Academy of Carbon Neutrality, Fujian Normal University, Fuzhou, 350117, China
[3]Department of Geography and Planning, University of Toronto, Ontario, ON M5S 3G3, Canada
[4]International Institute for Earth System Science, Nanjing University, Nanjing, 210023, China
[5]Key Laboratory of Ecosystem Network Observation and Modeling, Institute of Geographic Sciences and Natural Resources Research, Chinese Academy of Sciences, Beijing, 100101, China
[6]International Research Center of Big Data for Sustainable Development Goals, Beijing, 100094, China
[7]State Key Laboratory of Remote Sensing Science, Aerospace Information Research Institute, Chinese Academy of Sciences, Beijing, 100101, China
[8]Satellite Application Center for Ecology and Environment, Ministry of Ecology and Environment of the People's Republic of China, Beijing 100094, China
[9]College of Ecology and Environment, Hainan University, Haikou, 570228, China

*Correspondence to*: Xudong Lin (xudonglin50@gmail.com) and Jing M. Chen (jing.chen@utoronto.ca)

**Abstract.** Forest age is crucial for both carbon cycle modelling and effective forest management. Remote sensing provides crucial data for large-scale forest age mapping, but existing products often suffer from low spatial resolutions (typically 1,000 m), making them unsuitable for most forest stands in China, which are generally smaller than this threshold. Recent studies generated static forest age products for 2019 (CAFA V1.0) (Shang et al., 2023a) and 2020 (Cheng et al., 2024) at a 30-m spatial resolution. However, their low temporal resolution limits their applicability for tracking multi-year forest carbon changes. This study aims to generate China's annual forest age dataset (CAFA V2.0) at a 30-m resolution from 1986 to 2022, utilizing forest disturbance monitoring and machine learning techniques. Forest disturbance monitoring, which typically has lower uncertainty compared to machine learning approaches, is primarily employed to update annual forest age. The modified COLD (mCOLD) algorithm, which incorporates spatial information and bidirectional monitoring, was used for forest disturbance monitoring. For undisturbed forests, forest age was estimated using machine learning models trained separately for different regions and forest cover types, with inputs including forest height, vegetation indices, climate, terrain, and soil data. Additionally, adjustments were made for underestimations in the Northeast and Southwest regions identified in CAFA V1.0 using additional reference age samples and region-specific and forest type-specific models. Validation, using a randomly selected 30% of two reference datasets, indicated that the mapped age of disturbed forest exhibited a small error of ±2.48 years, while the mapped age of undisturbed forest from 1986 to 2022 had a larger error of ±7.91 years. The generated 30 m annual forest age dataset can facilitate forest carbon cycle modelling in China, offering valuable insights for national forest

management practices. The CAFA V2.0 dataset is publicly available at https://doi.org/10.6084/m9.figshare.24464170 (Shang et al., 2023b).

## 1 Introduction

Forest age, defined as the average age within a forest pixel (Shang et al., 2023a), is a key factor influencing both carbon sequestration and emission (Zhang et al., 2015) and determining long-term trends in forest carbon balance (Chen et al., 2000, 2003; He et al., 2012; Kurz and Apps, 1999; Zhang et al., 2013, 2014). Ignoring the influence of forest age on carbon sequestration can result in a 13-20% error in simulated forest carbon sinks, a discrepancy closely approximating the impact of climate change (Bellassen et al., 2011). Hence, accurately delineating forest age is essential for both effective forest management and precise carbon cycle modelling (Lin et al., 2023; Shang et al., 2023a).

Field measurements and surveys can accurately obtain forest age by extracting tree cores or utilizing records of the planting year. However, this method is time-consuming, labor-intensive, and costly when applied to large-scale forest age mapping (Lin et al., 2023). Moreover, the age measured for individual trees may not represent the average age of the entire stand, limiting its applicability for comprehensive forest age mapping (Besnard et al., 2021; Racine et al., 2014; Véga and St-Onge, 2008). In contrast, remote sensing provides continuous and repetitive observations of the Earth's surface, enabling large-scale forest age mapping (Vastaranta et al., 2016; Yang et al., 2020).

Many studies utilize remote sensing for forest age mapping, and the methods they used could be categorized into two types (Shang et al., 2023a): the forest height-based methods (shortened as "height-based method") and the forest disturbance monitoring-based methods (shortened as "disturbance-based method"). The height-based method begins by establishing a relationship between tree height and forest age using ground sample data, and then uses LiDAR tree height data to estimate forest age (Zhang et al., 2014, 2017). This relationship is often modelled using stand growth equations (Lin et al., 2023). Considering that forest growth is also influenced by environmental factors such as climate and terrain (Li et al., 2023; Lin et al., 2023), some studies employ machine learning techniques to construct intricate models using tree height, climate, and terrain data for forest age mapping (Besnard et al., 2021; Diao et al., 2020; Shang et al., 2023a; Zhao et al., 2021). However, current methods still exhibit significant uncertainties ranging from 12 to 48 years (Shang et al., 2023a). This is primarily due to the variability in the relationship between tree height and forest age, which not only correlates with climate and terrain factors but also varies with forest cover types (Lin et al., 2023). Therefore, incorporating forest cover type, alongside tree height, climate, and terrain considerations, may be an effective approach to improving the accuracy of forest age mapping.

The disturbance-based method primarily utilizes forest disturbance monitoring algorithms to identify the years when forests regrow after disturbances, estimating forest age accordingly (Xiao et al., 2023). Compared to the height-based method, the disturbance-based method typically results in lower uncertainties in estimating forest age (Shang et al., 2023a), but it is limited to estimating age within forest disturbance areas only. Commonly used forest disturbance monitoring algorithms include LandTrendr (Kennedy et al., 2010), CCDC (Zhu and Woodcock, 2014), and COLD (Zhu et al., 2020). Among these, CCDC

and COLD algorithms detect forest disturbances using all available time-series data, showing superior performance over LandTrendr, which relies on annual composite data for disturbance monitoring (Qiu et al., 2023). COLD, as an improved version of CCDC, enhances the accuracy of forest disturbance monitoring (Zhu et al., 2020). However, as a single-pixel time-series forest disturbance monitoring algorithm, COLD may omit forest disturbances within specific areas (Ye et al., 2023), potentially leading to significant biases in forest age estimation. Therefore, to improve the accuracy of forest age mapping, it is urgent to integrate spatial information into forest disturbance monitoring.

Several forest age products have been developed for China (Zhang et al., 2017, 2014; Xiao et al., 2023; Shang et al., 2023a; Cheng et al., 2024; Besnard et al., 2021). Early studies produced three sets of forest age products with a spatial resolution of 1000 meters for the years 2005 and 2010 using the height-based method (Zhang et al., 2014; Zhang et al., 2017; Besnard et al., 2021). However, the 1000-meter resolution averages forest stands within each pixel, leading to overestimations of young forests and underestimations of old forests. In recent years, driven by the demand for precise simulation of forest carbon dynamics and the availability of high-resolution remote sensing data, several high-resolution forest age products have been successfully generated. For example, Xiao et al. (2023) estimated forest age in disturbance areas across China at a 30-meter resolution in 2020 using the CCDC disturbance monitoring algorithm. Cheng et al. (2024) combined machine learning algorithms based on tree height, climate, and terrain with the LandTrendr disturbance monitoring algorithm to obtain forest age data for China in 2020. Our previous work (Shang et al., 2023a) utilized machine learning algorithms and the COLD disturbance monitoring algorithm to estimate nationwide forest age at a 30-m resolution in 2019 (CAFA V1.0). Compared to earlier products that used the height-based method alone, integrating it to estimate forest age with the disturbance-based method for updating forest age significantly enhances reliability. However, significant discrepancies still exist among current forest age products, which provide data for single years only, thus overlooking substantial changes in forest age before and after disturbances. These omitted changes can have a large impact on forest carbon modeling. When using single-year forest age data, process-based ecosystem models often underestimate the forest carbon uptake prior to the most recent forest disturbance and fail to account for the carbon release from multiple forest disturbances, leading to substantial uncertainties in forest carbon modeling (Zhang et al., 2025; Yu et al., 2020). In contrast, long-term forest age products can capture these carbon dynamics, making them more valuable for forest carbon modeling and forest management (Zhang et al., 2025; Chorshanbiyev et al., 2024). Therefore, it is urgent to generate long-term, high-resolution forest age products to support China's carbon neutrality researches (Besnard et al., 2021; Schumacher et al., 2020; Yu et al., 2020).

This study aims to generate China's annual forest age dataset (CAFA V2.0) from 1986 to 2022 at a 30-m resolution using the modified COLD (mCOLD) forest disturbance monitoring and a machine learning method. The forest age in 2019 was regarded as the baseline to update annual forest age, which was derived using the forest cover type specific machine learning models trained with inputs derived from forest height, vegetation indices, climate, terrain, and soil data. The mCOLD algorithm, which incorporates spatial information and bidirectional monitoring, was employed for forest disturbance monitoring from 1986 to 2022. The age of disturbed forests would be updated by the years since the disturbance. The mapped forest age would be validated using two reference datasets: one comprising 12,328 interpreted reference forest disturbance datasets in China,

and the other consisting of 5,304 forest field survey samples in China. The generated 30 m annual forest age dataset can facilitate forest carbon cycle modelling in China, offering valuable insights for national forest management practices.

## 2 Study area and data

### 2.1 Study area

The study area is China's forest region (Fig. 1), which has the largest afforested area globally. From 2000 to 2017, China accounted for 25% of the global net increase in leaf area, with forestation contributing 42% of this increase (Chen et al., 2019). China is typically divided into six regions (Shang et al., 2023a): North (N) region with forest coverage of 21.09%, Northwest (NW) region with forest coverage of 8.21%, Northeast (NE) region with forest coverage of 41.59%, South (S) region with forest coverage of 44.63%, East (E) region with forest coverage of 40.64%, and Southwest (SW) region with forest coverage of 25.75%. The forest covers in China can be broadly classified into five types: Evergreen Broad-leaved Forests (EBF), Deciduous Broad-leaved Forests (DBF), Evergreen Coniferous Forests (ENF), Deciduous Coniferous Forests (DNF), and Mixed Forests (MF). Deciduous forests dominate in the north regions of China, and evergreen forests dominate in the south regions of China. DBFs dominate in the N and NE regions, particularly in the provinces of Liaoning, Jilin, Heilongjiang, Hebei, and Shanxi (Zhang et al., 2021a). These areas have a climate characterized by cold winters and warm summers, creating favourable conditions for the growth of DBFs. EBFs are primarily distributed in the E and S regions, including the provinces of Zhejiang, Fujian, Jiangxi, Guangdong, Guangxi, and Hainan, where a warm, humid climate with abundant rainfall supports the growth of evergreen species. ENFs are concentrated in the region SW, including parts of Sichuan, Yunnan, and Tibet, where mountainous terrain, high elevations, and a cool, moist climate favour evergreen conifer species. DNFs are less common, primarily found in the Northeast, particularly in the Greater Khingan Mountains and the Lesser Khingan Mountains. MFs are also sparsely distributed, typically occurring in transitional zones or areas with complex ecological conditions, such as the Changbai Mountains in the Northeast and the Hengduan Mountains in the Southwest. These mixed forests, containing both coniferous and broad-leaved species, play a crucial role in maintaining ecosystem balance and supporting biodiversity.

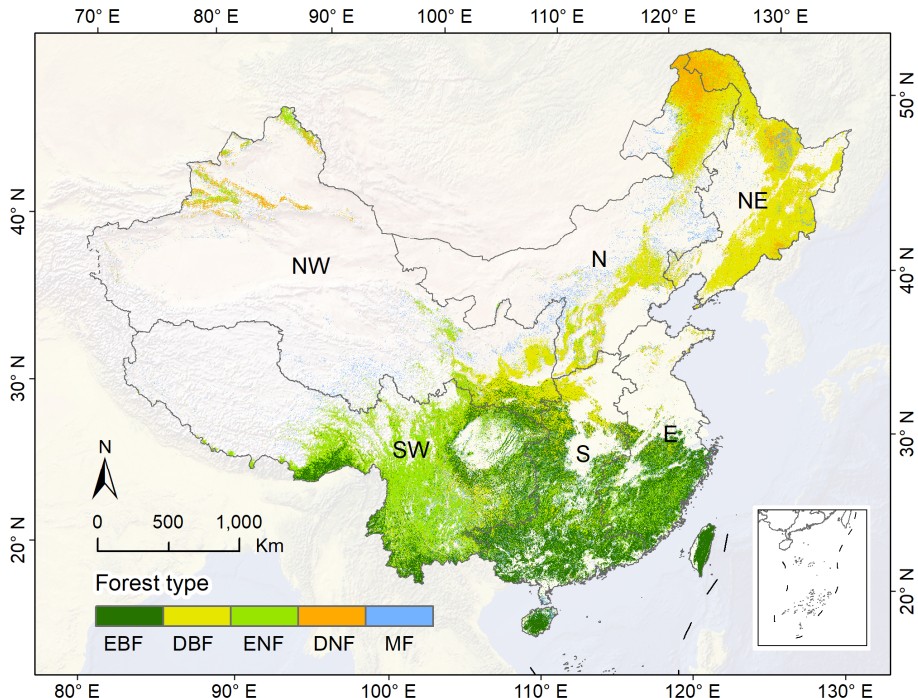

**Figure 1: Spatial distribution of China's forest cover types in 2019 and six regions.** The forest cover types were merged from three forest cover type products (Shang et al., 2023a): GLC_FCS30 (Zhang et al., 2021c) at the 30 m resolution, GLASS-GLC (Liu et al., 2021) at the 30 m resolution, and ESA CCI LC (ESA, 2017) resampled into the 30 m resolution from the 300 m resolution. EBF: evergreen broad-leaved forests, DBF: deciduous broad-leaved forests, ENF: evergreen coniferous forests, DNF: deciduous coniferous forests, MF: mixed forests. N: Northern region, NE: Northeast region, E: East China, S: South China, SW: Southwest region, NW: Northwest region. The map lines may contain disputed territories.

## 2.2 Data

### 2.2.1 Landsat data

Landsat Collection 2 Tier 1 surface reflectance data were used for forest age mapping in two parts: estimating the age of disturbed forests through forest disturbance monitoring, and modelling the age of undisturbed forests using machine learning methods combined with forest height data. For mapping the age of undisturbed forests or forests before disturbance, forest height data may not be available from existing China's forest height products (see Section 2.2.3 for details). In such cases, Landsat data should also be used to retrieve forest height.

This version of Landsat data includes all available images from Landsat 5-8 spanning from 1986 to 2022, featuring multi-spectral bands such as blue (B), green (G), red (R), near-infrared (NIR), shortwave infrared 1 (SWIR1), and shortwave infrared 2 (SWIR2), along with quality control (QA) bands. These bands are essential for forest disturbance monitoring, forest height, and age mapping. Pre-processing was conducted using the QA band (QA_PIXEL) by applying bitwise operations to identify clouds and shadows in the image according to the QA descriptions (Zhu et al., 2015), and filtering out pixels affected by clouds and shadows (Zhang et al., 2024). A time series filter (Shang et al., 2022) was also used for screening the outliers in the Landsat

time series data. For forest disturbance monitoring, the surface reflectance of the six spectral bands was employed using the mCOLD algorithm (Shang et al., 2025). In addition to surface reflectance, two vegetation indices, the Normalized Difference Vegetation Index (NDVI) and Near-Infrared Reflectance of Vegetation (NIRv), were utilized for forest height and age mapping. NDVI is an approximate indicator of vegetation greenness (Zeng et al., 2022; Zhu et al., 2016), while NIRv serves as an approximate indicator of vegetation productivity (Badgley et al., 2017; Shang et al., 2023a).

### 2.2.2 Reference samples

The reference forest age samples (Fig. 2a) were utilized for forest age mapping and validation of undisturbed forests from 1986 to 2022. This dataset comprises two components: 3,121 samples obtained from field surveys conducted under the Strategic Priority Project of Carbon Budget (SPPCB) project (Fang et al., 2018), and 2,183 samples derived from literature reviews (Luo et al., 2014; Cook-Patton et al., 2020). Each forest survey plot within SPPCB provides key attributes relevant to age mapping such as forest age, forest cover type, survey locations, and survey dates. For each region and forest type, 70% of the reference forest age samples were randomly chosen for model training, with the remaining 30% reserved for validation.

The reference forest disturbance samples (Fig. 2b) were used to validate the ages of forests disturbed at least once between 1986 and 2022. The age of these samples was derived from the number of years since the disturbance event. The reference forest disturbance samples were interpreted through analysis of time series images from Google Earth, PlanetScope, Sentinel-2, or Landsat 5/7/8, with each event confirmed by at least two clear-sky images taken before and after the disturbance (Qiu et al., 2023; Shang et al., 2025). The interpretation of reference forest disturbance samples was performed in three stages. Initially, samples were divided into sets of 1,000, with each of the 13 experts independently interpreting three sets. This ensured that each set was reviewed by at least three experts. For each set, samples unanimously identified by three experts (consistency rate 43%–81%) were accepted as final. In the second stage, samples identified by two experts were reviewed by a fourth expert, while those with no consensus were reviewed by both a fourth and fifth expert. Samples confirmed by at least three experts were accepted as final. In the final stage, remaining unconfirmed samples were voted on by all experts, with those receiving over 50% of the votes being accepted as final. A total of 12,328 forest disturbance samples were interpreted, with 4,168 samples having at least one forest disturbance. Of these, 2,157 samples experienced a single disturbance event between 1986 and 2022, 1,274 points had two disturbances, and 737 points were disturbed more than twice.

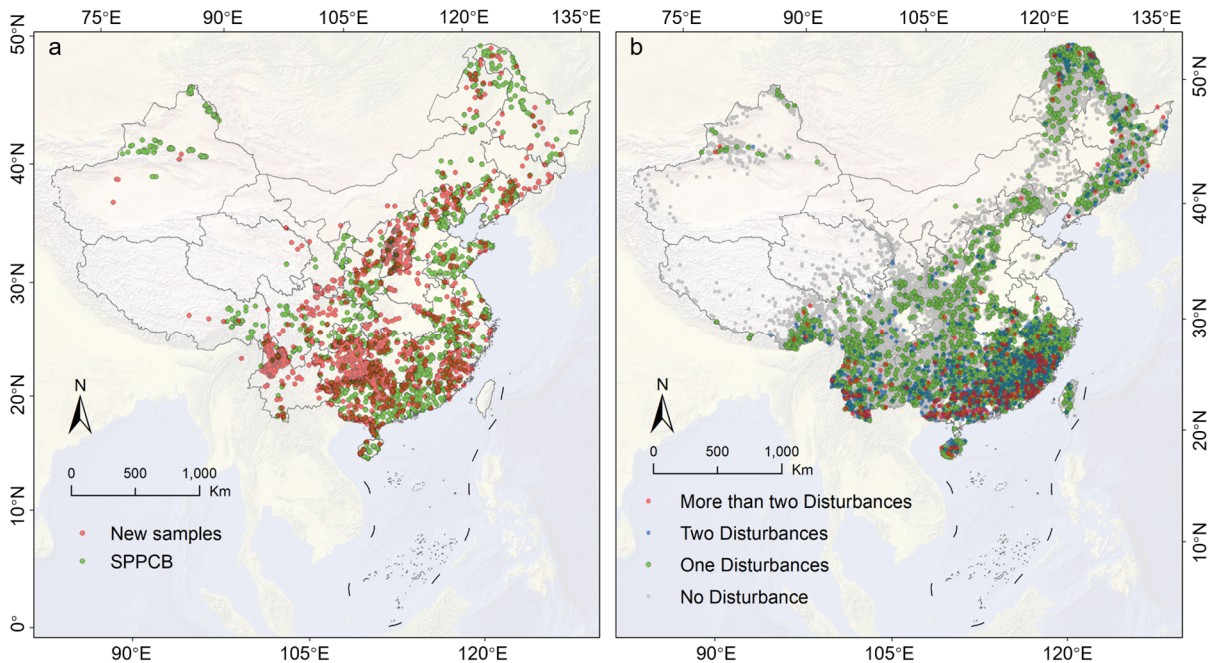

**Figure 2: Spatial distribution of reference samples for forest age mapping and validation. a** is the reference forest age samples, and **b** is the reference forest disturbance samples. The map lines may contain disputed territories.

### 2.2.3 Forest extent and forest cover type data

The China Land Cover Dataset (CLCD) (Yang and Huang, 2021) was used to indicate the dynamic forest extent of the forest age product. This dataset provides annual land cover information including forest cover extent for China from 1985 to 2022 at a 30 m spatial resolution, generated using Landsat imagery and random forest classifiers. It also had a comparable reliability to Hansen's Global Forest Change (GFC) dataset (Hansen et al., 2013) in terms of indicating forest changes (Yang and Huang, 2021). Several studies have also demonstrated that the CLCD offers higher accuracy than other land cover products across China (Zhang et al., 2022; Ji et al., 2024).

A merged forest cover type dataset (Fig. 1) was used for forest age mapping, as forest age mapping requires forest cover types as inputs, which were not provided by the CLCD product. This dataset was merged from three forest cover type products (Shang et al., 2023a): GLC_FCS30 from 1985 to 2022 (Zhang et al., 2021c) at the 30 m resolution, GLASS-GLC from 1985 to 2020 (Liu et al., 2021) at the 30 m resolution, and ESA CCI LC from 1992 to 2019 (ESA, 2017) resampled into the 30 m resolution from the 300 m resolution. There were four merging rules: first, a forest type was designated if at least two products identified the same forest cover type; second, if all three products had different types, the type from GLC_FCS30 was used, as it closely matched China's ninth forest resource report; third, if GLC_FCS30 indicated non-forest, the type from GLASS-GLC was used due to its higher spatial resolution than ESA CCI; last, if both GLC_FCS30 and GLASS-GLC indicated non-forest, the type from ESA CCI was utilized. The merged dataset and the three forest cover type datasets were validated against the field forest cover type data from the SPPCB project (Fang et al., 2018), and the accuracy of the merged dataset improved

significantly. Specifically, the Kappa coefficient of the merged dataset was 3.2% higher than that of GLC_FCS30, 6.31% higher than GLASS-GLC, and 8.4% higher than ESA CCI LC.

### 2.2.4 Forest height data

Two forest height products with the same forest definition at a 30-meter spatial resolution for the year 2019 (Potapov et al., 2021; Liu et al., 2022) were employed to map the age of undisturbed forests. Potapov et al. (2021) utilized machine learning methods with Landsat data and Global Ecosystem Dynamics Investigation (GEDI) footprint forest height data to generate a global forest canopy height map at a 30-meter spatial resolution for the year 2019 (shortened to Potapov's forest height product), while Liu et al. (2022) developed a neural network guided interpolation (NNGI) method to derive China's forest height map at 30-meter spatial resolution for 2019 (shortened to Liu's forest height product), using Landsat data along with GEDI and ICESat-2 footprint forest height data. Due to consideration of topographic influences and high-quality control standards, Liu's forest height product exhibited higher accuracy but had a smaller forest extent in China than Potapov's forest height product (Liu et al., 2022). Therefore, this study primarily used Liu's forest height product. When Liu's product was missing compared with the forest extent identified in CLCD, Potapov's forest height product was used as a substitute.

For forest pixels with missing forest heights from both Liu's and Potapov's two products (0.32% of pixels, based on the forest extent in CLCD), forest height was estimated using a machine learning method (detailed in Section 3.2.2) that integrates Landsat data, climate data, terrain data, and GEDI footprint forest height data. The input Landsat data consists of surface reflectance from Landsat 5, 7, and 8 and two calculated vegetation indices (NDVI and NIRv). The input data from GEDI, launched by NASA in December 2018 and covering the Earth's land surface from 51.6°N to 51.6°S (Dubayah et al., 2020), primarily includes the L2A relative height data, which has demonstrated the best performance in global forest height mapping (Potapov et al., 2021).

### 2.2.5 Other data

The vegetation, terrain, climate, and soil data used for mapping forest age were summarized in Table 1. Aside from forest height and cover type data, vegetation data also includes two vegetation indices, NDVI and NIRv, calculated from Landsat data. Terrain data mainly includes the slope and aspect calculated from 30-meter resolution NASA DEM data, chosen for its extensive use as one of the most prevalent global DEM products across diverse applications (Su et al., 2015). Climate data includes temperature and precipitation, which are further divided into eight variables: highest annual temperature (HAT), lowest annual temperature (LAT), mean annual temperature (MAT), annual temperature range (ATR), highest annual precipitation (HAP), lowest annual precipitation (LAP), mean annual precipitation (MAP), and annual precipitation range (APR). These variables are all relevant to forest growth, as they influence various aspects of forest health and development, such as tree physiology, species distribution, and overall ecosystem productivity (Leuschner and Ellenberg, 2017; Chapin et al., 2011). These climate variables were extracted from a dataset of monthly precipitation and temperature at a 1-kilometer resolution for China spanning the period from 1901 to 2023 (Peng, 2020). This dataset was derived from global CRU and

WorldClim climate datasets and data from 496 observation stations using the Delta space downscaling method (Peng, 2020). Soil type data was obtained from the SoilGrids 2.0 product, which applies machine learning to soil observations from approximately 240,000 global locations, using over 400 environmental covariates related to vegetation, terrain, climate, geology, and hydrology to map global soil properties at a 250-meter resolution (Poggio et al., 2021).

**Table 1: Descriptions of the vegetation, terrain, climate, and soil data used for forest age mapping.**

| Data type | Data name | Resolution | Years | Data sources | References |
|---|---|---|---|---|---|
| Vegetation | Forest height map | 30m | 2019 | Published data | Liu et al. (2022); Potapov et al. (2021) |
| | Forest height | 30m | 1986-2019 | GEDI and Landsat data | This study |
| | NDVI | 30m | 1986-2019 | Landsat data | This study |
| | NIRv | 30m | 1986-2019 | Landsat data | This study |
| Terrain | Slope | 30m | 2020 | NASA DEM | Uuemaa et al. (2020) |
| | Aspect | 30m | 2020 | NASA DEM | Uuemaa et al. (2020) |
| Climate | Highest annual temperature | 1km | 1986-2022 | Published data | Peng (2020) |
| | Lowest annual temperature | 1km | 1986-2022 | Published data | Peng (2020) |
| | Mean annual temperature | 1km | 1986-2022 | Published data | Peng (2020) |
| | Annual temperature range | 1km | 1986-2022 | Published data | Peng (2020) |
| | Highest annual precipitation | 1km | 1986-2022 | Published data | Peng (2020) |
| | Lowest annual precipitation | 1km | 1986-2022 | Published data | Peng (2020) |
| | Mean annual precipitation | 1km | 1986-2022 | Published data | Peng (2020) |
| | Annual precipitation range | 1km | 1986-2022 | Published data | Peng (2020) |
| Soil | Soil Type | 250m | 2021 | SoilGrids (V2.0) | Poggio et al. (2021) |

## 3 Methods

### 3.1 Forest age mapping

Forest age mapping was divided into two parts: disturbed forests and undisturbed forests. For disturbed forests, age was primarily determined based on forest disturbance monitoring. For undisturbed forests, random forests were used to map forest age based on vegetation, terrain, climate, and soil data. Figure 3 shows the flowchart of forest age mapping, and there were three steps: mapping the age of disturbed forests through forest disturbance monitoring (section 3.2.1), forest height retrieval for undisturbed forests (section 3.2.2), and mapping the age of undisturbed forests through random forests (section 3.2.3).

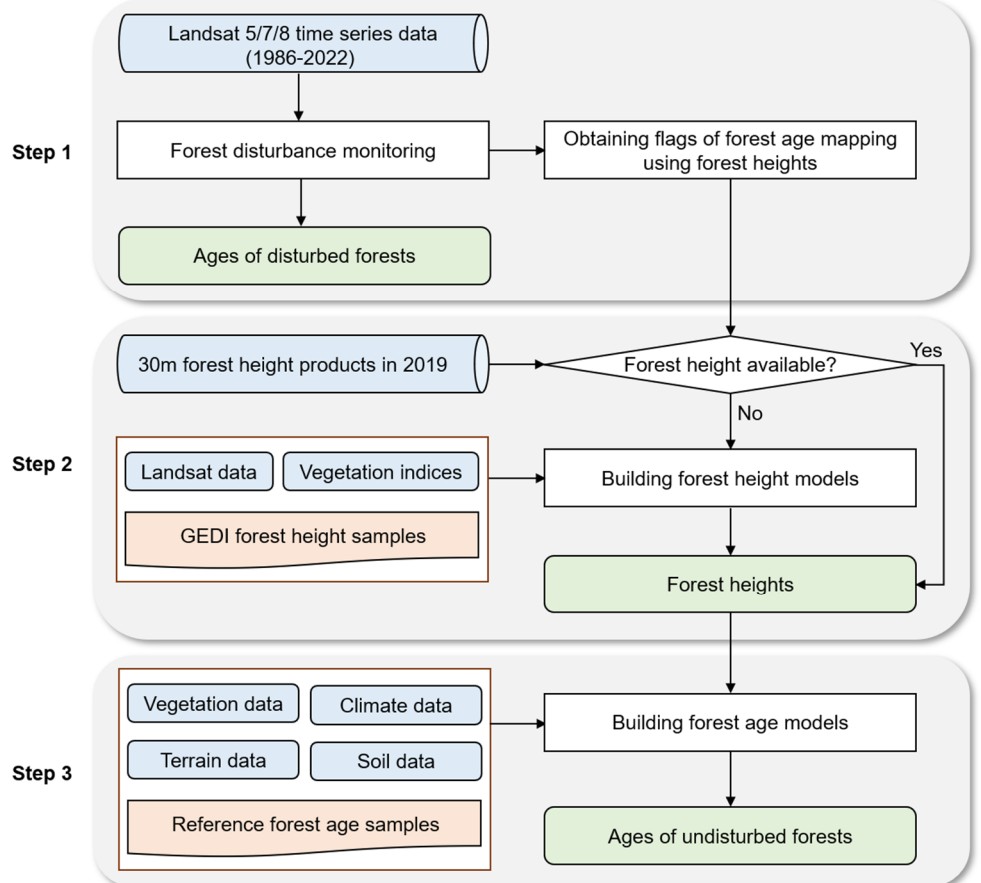

**Figure 3: The flowchart of China's annual forest age mapping from 1986 to 2022.** There were three steps: mapping the age of disturbed forests through forest disturbance monitoring (Step 1), forest height retrieval for undisturbed forests (Step 2), and mapping the age of undisturbed forests through random forests (Step 3).

Annual updates of forest age were based on the forest age in 2019 and the dates of detected forest disturbance (Fig. 4), with the age resetting to 0 in the disturbance year. For areas with no forest disturbance between 1986 and 2022 (Fig. 4a), the age for each year can be updated based on the number of years since 2019. For areas with a forest disturbance after 2019 (Fig. 4b), the age for years before 2019 can be updated based on the number of years since 2019, while the age for years after disturbance can be updated based on the number of years since the disturbance year. For areas with one (Fig. 4c) or more than one occurrence of forest disturbance (Fig. 4d) before 2019, the age for years before the first forest disturbance should be mapped based on the retrieved forest height and other data using random forests, while the age for years after each disturbance can be updated based on the number of years since the disturbance year.

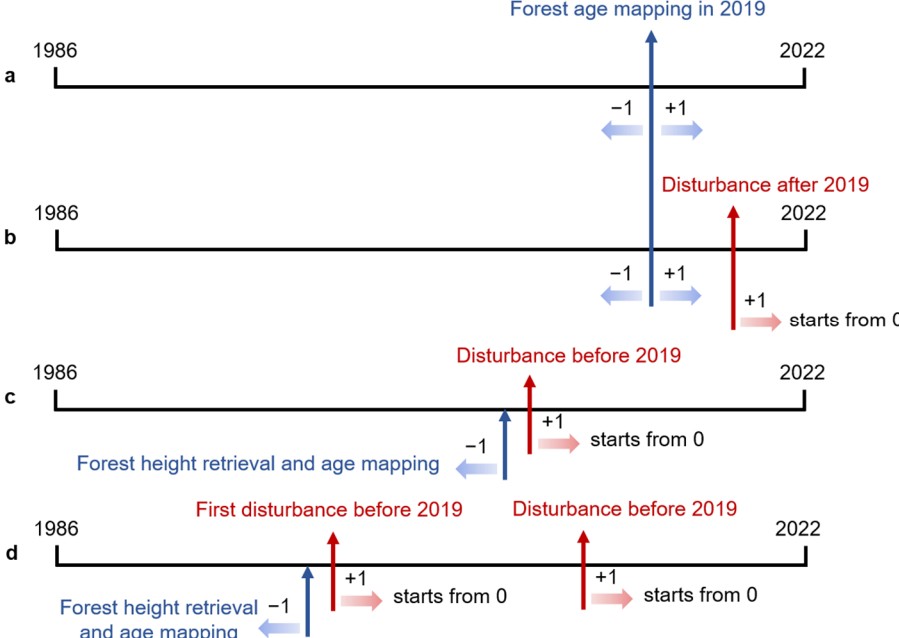

**Figure 4: Conceptual diagram of annual forest age updates. a** is for the situation with no forest disturbance between 1986 and 2022; **b** is for the situation with a forest disturbance after 2019; **c** is for the situation with a forest disturbance before 2019; **d** is for the situation with more than one forest disturbance before 2019. If there is at least one disturbance before 2019, the forest height before the first disturbance should be retrieved to map forest age.

### 3.1.1 Mapping the age of disturbed forests through forest disturbance monitoring

Forest disturbances were detected using the mCOLD algorithm (Shang et al., 2025). The original COLD algorithm employed a time series model to predict multi-band surface reflectance (Zhu et al., 2020). It then used the chi-square distribution to quantify the differences between the multi-band predictions and observations, referred to as the change magnitude. Forest disturbances were confirmed based on two criteria (Zhu et al., 2020): a change magnitude criterion (a change magnitude calculated from the chi-square distribution corresponding to 0.99) and a timing criterion (at least six observations).

There were three modifications in mCOLD. First, spatial information was incorporated into forest disturbance monitoring. Due to differences in forest types and locations, adjacent pixels may exhibit varying change magnitudes from the same disturbance, and the algorithm with a uniform standard for disturbance confirmation may result in omissions. To avoid the omissions, the data-adapted iterative steering kernel regression (DISKR) algorithm (Takeda et al., 2007) was used to incorporate spatial information from adjacent pixels within a window to enhance forest disturbance monitoring (Shang et al., 2025).

Second, the timing criterion for confirming a forest disturbance was revised from requiring at least six observations to meeting both a minimum of six observations and a minimum disturbance duration. This adjustment was made to account for the varying density of valid Landsat time series observations across different locations, as data density can be twice as high in overlapping areas compared to non-overlapping areas (Zhang et al., 2021b). Relying solely on the number of observations

could lead to inconsistencies in the timing of detected disturbances. By incorporating the duration criterion, this modification could address potential discrepancies and improve the temporal consistency of detected forest disturbances (Shang et al., 2022).

Last, forest disturbance monitoring was revised from a unidirectional approach (tracking from past to present) to a bidirectional approach (incorporating both past-to-present and present-to-past tracking). In unidirectional tracking, detecting early forest disturbances was often less accurate than detecting later ones due to the typically fewer observations available for building a reliable time series to detect the early disturbance (Zheng et al., 2022). Bidirectional monitoring addresses this issue by transforming early disturbances in the past-to-present tracking into later disturbances in the present-to-past tracking. This method ensured sufficient data for developing a more accurate time series for both early and late disturbances, thereby enhancing the overall accuracy of disturbance monitoring (Shang et al., 2025).

### 3.1.2 Forest height retrieval for undisturbed forests

Random forests with boosting trees, which can minimize biases and improve overall model performance (Jahan et al., 2021), were employed to estimate forest height in areas where forest height data was unavailable. The GEDI footprint forest height served as the training dataset for the forest height models, with the screening process following the methodology outlined by Liu et al. (2022). Specifically, for each GEDI footprint, six relative heights including RH75, RH80, RH85, RH90, RH95, and RH100 were selected. To reduce the influence of outliers, the maximum and minimum values were excluded, and the average of the remaining four relative height values was used to represent the relative height for each footprint. Furthermore, the quality of the samples was further ensured by filtering them using elevation data. TanDEM-X and SRTM elevation values recorded in the GEDI dataset were compared, and any samples with an elevation difference greater than 10 meters were discarded.

To reduce the discrepancy between the retrieved forest height and the 2019 forest height products, the input factors for the tree height model were based on the study by Liu et al. (2022), incorporating DEM, slope, aspect, temperature, precipitation, and NDVI data. Additionally, this study expanded the input factors by incorporating Landsat 8 surface reflectance and the calculated NIRv vegetation index, which approximates forest productivity (Badgley et al., 2019). This resulted in a total of 13 input factors used in the tree height model. The models were constructed separately for different environmental conditions and forest types across China. Specifically, six regions (East, South, North, Northeast, Northwest, and Southwest) and five forest types (EBF, ENF, DBF, DNF, and MF) were considered. Each model was trained using 70% of the filtered GEDI footprint forest height samples, with the remaining 30% used for validation. Since no GEDI footprint forest height samples were available prior to 2019, the forest height retrieval for years before 2019 at a specific forest type and region utilized the same model as 2019 but with varied inputs corresponding to their respective years. These forests, which required height retrieval, covered only a small portion of the total forest pixels, with an average of 0.42% (see details in Section 5.2).

In addition to estimating forest height in areas where forest height data was unavailable, forest height was also estimated for randomly selected samples in regions where the 2019 forest height product was available. This allowed for the evaluation of the differences between the estimated and product-based forest heights and their impact on forest age mapping (see details in Section 5.2).

### 3.1.3 Mapping the age of undisturbed forests through random forests

Random forests with boosting trees (Jahan et al., 2021) were also selected to map the age of undisturbed forests. This machine learning method showed the highest overall accuracy in forest age mapping at a 30 m spatial resolution among five stand growth equations and twelve machine learning methods (Lin et al., 2023). The model used fifteen inputs (Table 1): vegetation factors (forest height, NDVI, and NIRv), terrain factors (slope and aspect), climate factors (HAT, LAT, MAT, ATR, HAP, LAP, and MAP), and one soil factor (soil type). Tree height was selected due to its dominant role in forest age mapping (Lin et al., 2023), while NDVI and NIRv (Equations (1) and (2)) could reflect forest greenness and productivity. Terrain factors such as slope and aspect also affected forest growth (Lang et al., 2010). Temperature and precipitation were included as forest growth is sensitive to climatic conditions (Besnard et al., 2021). Soil type was also considered for its effect on vegetation growth.

$$NDVI = \frac{NIR_{Ref} - RED_{Ref}}{NIR_{Ref} + RED_{Ref}} \tag{1}$$

$$NIR_V = NIR_{Ref} * NDVI \tag{2}$$

where, $NIR_{Ref}$ represents the surface reflectance at the near-infrared band, and $RED_{Ref}$ represents the surface reflectance at the red band.

Given the regional and forest cover type heterogeneity across China, this study divided the model construction into six regions and five forest cover types. An RF model was developed using MATLAB for each forest cover type within each region. In regions where certain forest cover types had fewer than 30 sample points, those regions were merged, resulting in a total of 22 forest age models. This stratified approach aims to enhance the accuracy of forest age mapping as forest types can also influence the accuracy of forest age mapping (Lin et al., 2023). For each forest cover type within a region, 70% of the randomly selected reference forest age samples were used for model training, while the remaining 30% were used for validation. To mitigate the effects of autocorrelation between the input factors, this study proposed an automatic iterative selection mechanism, aimed at reducing autocorrelation by minimizing the number of input factors. The process began by using all input factors to estimate forest age, followed by extracting the contribution weights of each factor. Factors with a contribution weight of less than 0.5% were removed. The remaining factors were then used to re-estimate forest age. This iterative process was repeated three times, with factors contributing less than 0.5% being excluded in each iteration, and the final model after three iterations was used for forest age estimation. In addition to input feature screening, we also performed sensitivity analysis to determine the optimal thresholds for the minimum leaf size and number of trees for each model. The minimum leaf size was varied from 1 to 30, with an interval of 5, while the number of trees was adjusted from 50 to 300, with an interval of 50. The optimal thresholds were identified as those corresponding to the minimum RMSE of the mapped forest age (Table 2).

The absolute mean SHapley Additive exPlanations (SHAP) values were calculated to indicate the importance of each input factor, as SHAP values were widely to explain machine learning models (Lundberg and Lee, 2017; Lundberg et al., 2018). By evaluating the contribution of each input factor to the model, SHAP can rank the importance of these factors. The higher the

SHAP value, the greater the contribution of the factor to the model (Sun et al., 2023). The importance of each input factor for forest age mapping would be discussed in Section 5.1.

**Table 2: Parameters of the forest age mapping models for different regions and forest cover types.** EBF: evergreen broad-leaved forest, DBF: deciduous broad-leaved forest, ENF: evergreen coniferous forest, DNF: deciduous coniferous forest, MF: mixed forest. N: Northern region, NE: Northeast region, E: East China, S: South China, SW: Southwest region, NW: Northwest region.

| Models (region and forest type) | Minimum leaf size | Number of trees | Number of features |
|---|---|---|---|
| NW-DNF | 5 | 150 | 13 |
| NW-ENF | 5 | 150 | 14 |
| NW-DBF | 5 | 150 | 14 |
| NW-EBF/MF | 5 | 100 | 14 |
| SW-DNF/MF | 5 | 200 | 15 |
| SW-ENF | 5 | 100 | 12 |
| SW-DBF | 10 | 100 | 15 |
| SW-EBF | 10 | 150 | 15 |
| S-DNF/MF | 20 | 100 | 15 |
| S-ENF | 5 | 100 | 14 |
| S-DBF | 5 | 150 | 14 |
| S-EBF | 10 | 100 | 14 |
| E-DNF/MF | 20 | 50 | 14 |
| E-ENF | 10 | 100 | 15 |
| E-DBF | 10 | 100 | 14 |
| E-EBF | 10 | 100 | 15 |
| NE-DNF | 20 | 100 | 13 |
| NE-DBF | 20 | 100 | 15 |
| NE-EBF/ENF/MF | 5 | 200 | 14 |
| N-DNF | 10 | 200 | 14 |
| N-DBF | 5 | 150 | 14 |
| N-EBF/ENF/MF | 5 | 150 | 15 |

## 3.2 Validation methods

This study used the coefficient of determination ($R^2$) and Root Mean Square Error (RMSE) as two validation metrics (Lin et al., 2023) to reflect the accuracy of the mapped forest age.

$$RMSE = \sqrt{\frac{1}{N}\sum_{i=1}^{N}(X_i - X'_i)^2} \qquad (3)$$

$$R^2 = 1 - \frac{\sum_{i=1}^{N}(X_i - X'_i)^2}{\sum_{i=1}^{N}(X_i - \bar{X})^2} \qquad (4)$$

where, N represents the number of reference samples, i represents the i[th] sample, $X_i$ represents the forest age of the reference sample, $X'_i$ represents the mapped forest age, $\bar{X}$ represents the average value of the reference forest age. Higher $R^2$ values and lower RMSE values indicate better accuracy of forest age mapping. RMSE was considered a more indicative measure than $R^2$ because, while $R^2$ shows the degree of data dispersion and can be high even with large deviations, RMSE directly measures the deviation between mapped and reference forest ages, providing a clearer indication of the accuracy (Lin et al., 2023).

The mapped forest ages were validated in two parts: disturbed forests and undisturbed forests. For disturbed forests, the validation was performed using 30% of randomly selected reference forest disturbance samples (Fig. 2a); for undisturbed forests, validation was based on 30% of randomly selected reference forest age samples (Fig. 2b). In addition to nationwide validation, we specifically assessed the mapped forest ages in the NE and SW regions, where significant improvements were made in forest age mapping in the CAFA V2.0 product. Furthermore, since the age of undisturbed forests was mapped using separate RF models for different forest types, Section 5.3 would discuss the accuracy differences in forest age mapping among different forest cover types.

## 3.3 Uncertainty analysis

The uncertainty analysis primarily focused on the mapped ages of undisturbed forests in 2019 using the height-based method, as the disturbance-based method generally has lower uncertainties in mapping forest ages compared to the height-based method (Shang et al., 2023a). The uncertainties of the mapped forest ages mainly stem from the models and their inputs. Among the model inputs, forest height has the most significant impact on forest age mapping (see section 5.1 for details). Therefore, we concentrated solely on the uncertainty of input forest height in forest age mapping. The evaluation of the differences between the estimated and product-based forest heights and their impact on forest age mapping is discussed in section 5.2. To assess the uncertainties of the age mapping models, we kept the inputs constant while varying the forest heights estimated by forest stand growth equations. Based on Zhang's forest stand growth equations in China (Zhang et al., 2014), we calculated the relative forest heights for the years 2017, 2018, 2020, and 2021, according to the region and forest types. These forest heights were then used as inputs to map forest ages, and their standard deviation was calculated to represent the uncertainties.

## 4 Results

### 4.1 China's annual forest age at 30m resolution from 1986 to 2022

A dynamic forest age dataset (CAFA V2.0) covering the entire China from 1986 to 2022 (Shang et al., 2023b), with a spatial resolution of 30 m, was generated by integrating forest disturbance mapping and random forests methods. Figure 5 illustrates the distribution of forest ages for the year 2019, alongside comparisons with data from 1986, 2000, 2010, and 2022. This forest age dataset indicates that China's forest age structure predominantly consists of young and middle-aged forests, with an average forest age of $58.1 \pm 7.3$ years in 2019. Old forests were predominantly found in the northeast, northwest, and southwest regions of China. These areas, characterized by high mountains and minimal human interference, were largely comprised of natural and secondary forests. In contrast, forests disturbed at least once during the period from 1986 to 2022 exhibited younger ages, generally below 37 years. Such forests were mainly concentrated in the southeast and central southern regions, where human disturbances were more prevalent. Furthermore, in the northeast, there were also young forests that had regenerated after extensive forest fires, such as the devastating forest fire that occurred on May 6, 1987 (Cahoon Jr. et al., 1991). This fire caused varying degrees of damage across a vast area within China, impacting more than one million hectares of forests.

According to the 2020 map of China's planted and natural forests (Cheng et al., 2024), natural forests were older than planted forests, with average ages of $69.9 \pm 12.8$ years and $48.4 \pm 6.9$ years, respectively. Regional variations were evident, with the age gap between natural and planted forests ranging from 3.5 to 20.2 years. Southwest China had the oldest natural forests ($91.1 \pm 21.6$ years) and planted forests ($74.8 \pm 18.1$ years), while East and South China showed lower average ages due to higher disturbance frequencies.

The temporal dynamics of China's forest ages were primarily influenced by both forest loss disturbances (such as forest fire, harvest, and other disturbances) and forest gain disturbances (such as afforestation and reforestation), which mainly led to a reduction in China's average forest age. From 1987 to 2022, the age reduction caused by forest disturbances showed a decreasing trend, with an average age reduction of $-0.105 \pm 0.027$ years. However, in 1987, 2008, and 2021, the forest age reduction caused by disturbances was significant, indicating that there were more forest disturbances in these three years.

The uncertainty analysis was performed on the mapped ages of undisturbed forests in 2019 (Fig. 6). In most regions, the mapped forest age in 2019 exhibited relatively low uncertainty, with an average uncertainty of 8.7 years across China. However, the southwest region displayed higher uncertainty, exceeding 40 years in Tibet and certain mountainous areas of Sichuan province. This elevated uncertainty may be attributed to the heightened sensitivity of age mapping models to forest height in the southwest region (see section 5.1 for details). Additionally, the significant increase in forest height with age, as described by the forest stand growth equations in these areas (Zhang et al., 2014), further contributed to the increased uncertainty. Despite these regional variations, the mapped forest age in 2019 was generally stable and characterized by small uncertainties.

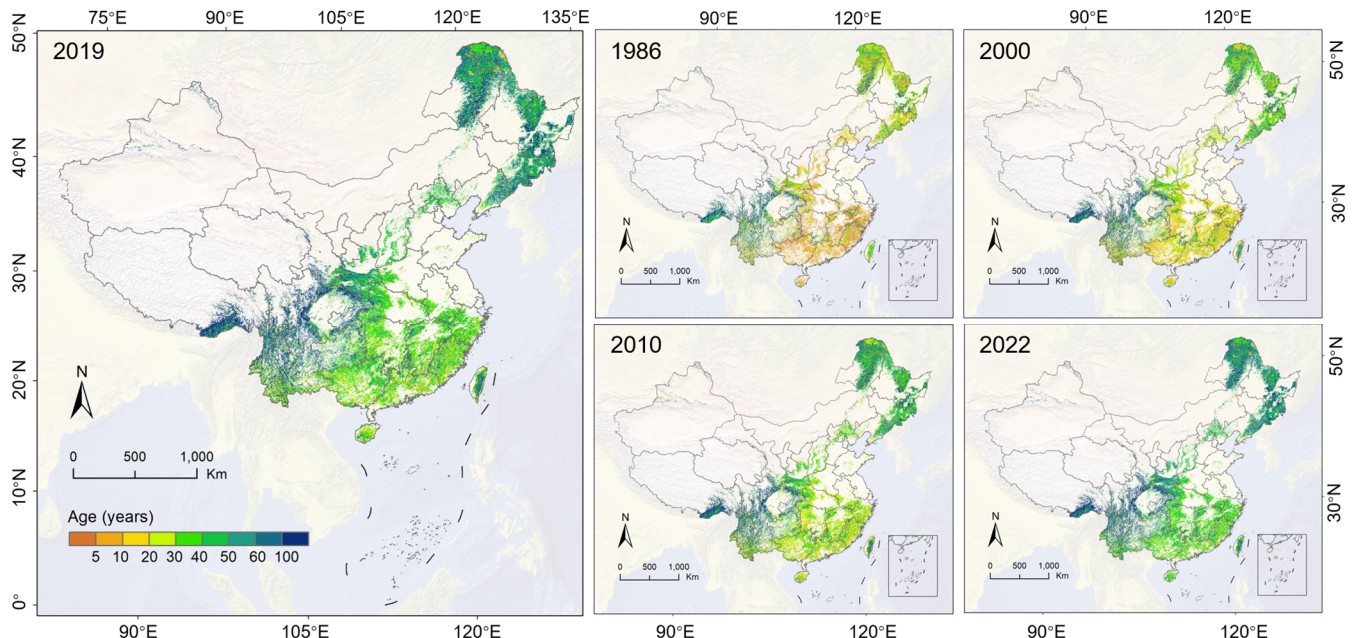

**Figure 5: Spatial distribution of China's forest age in 2019 and other selected years (1986, 2000, 2010, and 2022) in the CAFA V2.0 dataset.** The map lines may contain disputed territories.

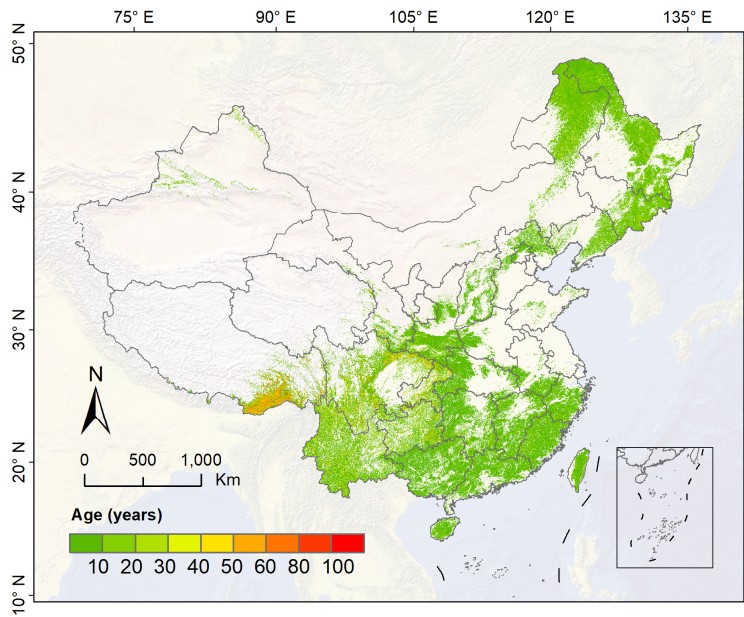

**Figure 6: Spatial distributions of the uncertainty of mapped forest age in 2019.** The map lines may contain disputed territories.

## 4.2 Validation of the forest age maps

The mapped forest age in 2019 was validated using 30% of two separate reference datasets (Fig. 7): one comprising 12,328 interpreted reference forest disturbance datasets and the other consisting of 5,304 forest field survey samples in China. For undisturbed forests, the field-surveyed age was transformed from the survey year to 2019 by adding the difference in years. For disturbed forests, the reference age in 2019 was determined by calculating the number of years since the last disturbance. Validation results indicated that the mapped age of disturbed forest exhibited a small error of ±2.48 years, while the mapped age of undisturbed forest from 1986 to 2022 had a relatively large error of ±7.91 years. Compared to version 1.0, the RMSE of CAFA V2.0 forest age for disturbed forests decreased by 1.15, and for undisturbed forests, the RMSE decreased by 0.49.

The enhancement in age accuracy for disturbed forests stemmed primarily from refining the forest disturbance monitoring algorithm. Figure 8 presents two typical examples of the mapped forest ages from forest disturbance monitoring using COLD, LandTrendr, and mCOLD. In both cases, mCOLD mapped a more accurate extent of the forest disturbances, and its results were used to mask the outputs of other algorithms. The first example pertains to a forest fire that occurred in 2010. Both COLD and LandTrendr detected the disturbance, but their detected extents were inaccurate with significant omissions. This resulted in substantial overestimations of forest age which was derived from the height-based methods. The second example involves a forest fire that took place in the winter. COLD identified the disturbance with an incomplete extent, while LandTrendr missed the disturbance entirely. Consequently, significant overestimations existed in the forest age products derived from these two algorithms. In comparison, mCOLD demonstrated superior performance in identifying the full extent of the forest disturbances compared to LandTrendr and COLD. As a result, forest age mapping accuracy improved notably for disturbed forests using mCOLD, particularly in the northeast and southwest regions of China, where the RMSEs decreased by 0.71 and 1.9, respectively (Fig. 9a and Fig. 9c).

The improvement in age accuracy for undisturbed forests was achieved through separate modelling for different regions and forest cover types, as well as enhancements in the reference forest age samples used for model training. In version 1.0, forest ages tended to be underestimated in the northeast and southwest regions of China due to limited reference samples, which were substantially improved in CAFA V2.0. As depicted in Fig. 9, the RMSEs decreased by 3.21 and 1.96, respectively (Fig. 9b and Fig. 9d). In conclusion, CAFA V2.0 represents a significant enhancement in the accuracy of forest age mapping across China.

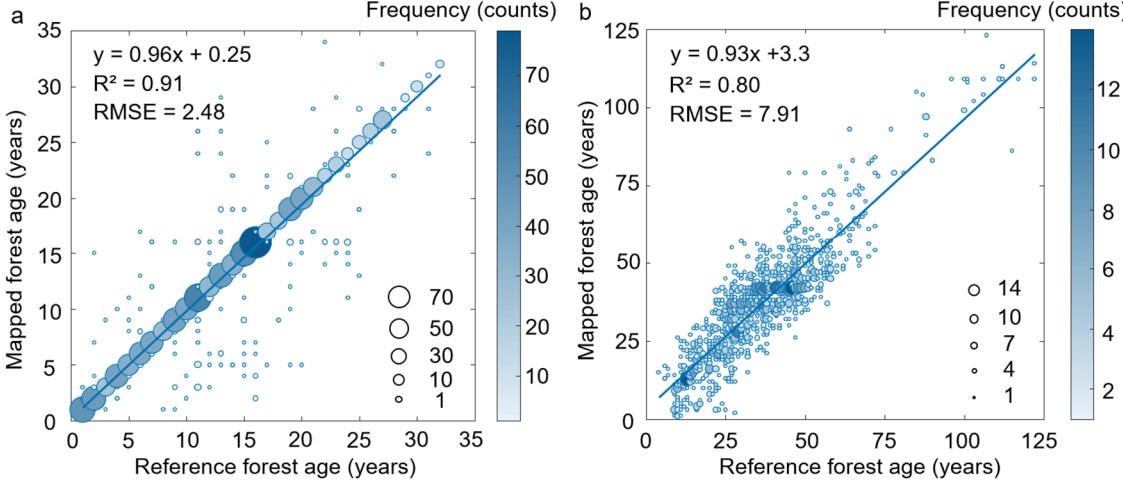

**Figure 7: Validation of China's forest age mapping in the CAFA V2.0 dataset. a** is for the age validation of forests disturbed at least once from 1986 to 2022, and **b** is for the age validation of undisturbed forests. The size and color of the circle represent the number of samples at that location.

**Figure 8: Typical examples of the mapped forest age in 2019 using forest disturbance monitoring.** The first example is a forest fire that occurred in March to June, 2010, and the second example is a forest fire that happened in January, 2017. The first column is Landsat RGB composited images after forest disturbance on March 10, 2010 (**a**) and February 8, 2017 (**e**). The second column (**b** and **f**) is the mapped forest age using COLD (Shang et al., 2023a). The third column (**c** and **g**) is the mapped forest age (calculated by deducting one year from the 2020 forest age) using LandTrendr (Cheng et al., 2024). The fourth column (**d** and **h**) is the mapped forest age using mCOLD.

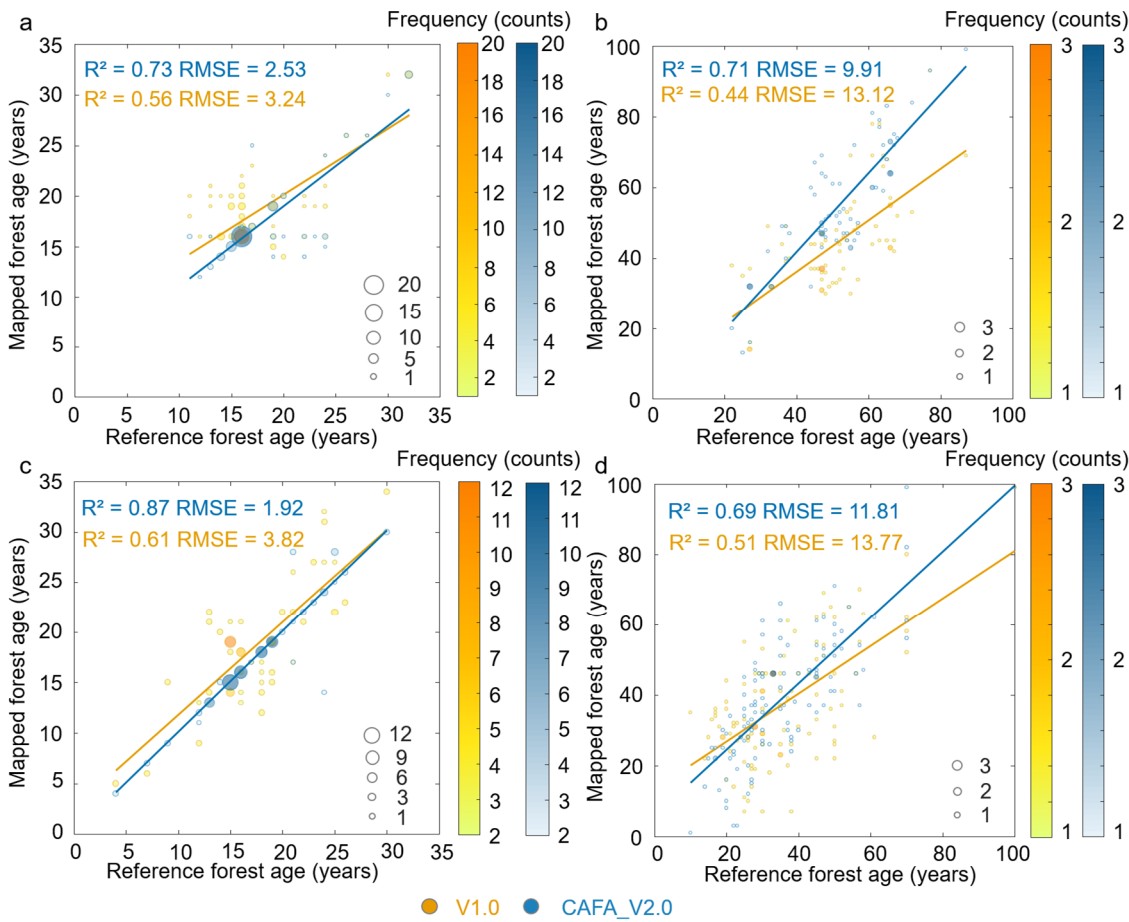

**Figure 9: Comparison of forest age mapping in the Northeast and Southwest regions between the V1.0 and V2.0 forest age products.**
**a** and **b** are for the Northeast region; **c** and **d** are for the Southwest region. **a** and **c** are comparisons of forest age mapping with at least one disturbance occurred from 1986 to 2022; **b** and **d** are comparisons of forest age mapping without disturbances between 1986 and 2022.

### 4.3 Comparisons with previous forest age products

The dynamic forest age dataset (CAFA V2.0) was compared with four existing static forest age products, including two at 1000 m resolution for the years 2005 (Age2005) (Zhang et al., 2014) and 2010 (Age2010) (Besnard et al., 2021), as well as two at 30 m resolution for the years 2019 (Age2019) (Shang et al., 2023a) and 2020 (Age2020) (Cheng et al., 2024). Figure 10 illustrates the spatial distributions of these four forest age products and the age differences between them and the CAFA V2.0 product. Overall, the spatial patterns of forest age in CAFA V2.0 align with the other four products, with older forests predominantly located in the northeast, northwest, and southwest regions of China. Significant forest age differences (≥ 30 years) are also observed in these regions, especially in areas affected by forest disturbances. The average forest age was 43.1 ± 9.5 years (within the 95% confidence interval) for Age2005, while the corresponding value for CAFA V2.0 in 2005 was 45.8 ± 8.1 years (Fig. 11a). For 2010, Age2010 had an average forest age of 45.3 ± 6.2 years, while CAFA V2.0 showed an age of 49.6 ± 7.9 years (Fig. 11b). For 2019, Age2019 had an average forest age of 53.2 ± 8.3years, while CAFA V2.0 showed

an age of 58.1 ± 7.3 years (Fig. 11c). Similarly, for 2020, the average ages for Age2020 were 57.3 ± 10.1 years, compared to 59.2 ± 8.5 years for CAFA V2.0 (Fig. 11d).

The four existing static forest age products were also validated using the same 30% reference forest age samples (Fig. 12). It should be noted that Age2005, Age2010, and Age2020 lacked forest age values for certain reference samples. These samples were displayed in Fig. 11 with a forest age of 0, but were excluded from the fitting line and the calculations of $R^2$ and RMSE. All four forest age products showed higher RMSE values compared to the CAFA V2.0 product, including our previous static forest age product, Age2019, which had an RMSE of 10.29 years. Age2020 exhibited the highest RMSE at 19.31 years, likely

due to young forests impacted by disturbances that were omitted by the LandTrendr forest disturbance monitoring algorithm (Qiu et al., 2023) but inaccurately classified as old forests with high ages in Age2020. Except for those disturbed forests, Age2020 demonstrated reasonable accuracy in forest age mapping. The CAFA V2.0 forest age product had the lowest RMSE of 7.91 years, owing to the modified forest disturbance monitoring and improvements in age mapping of undisturbed forests, particularly in the Northeast and Southwest regions, where additional reference samples and region- and forest cover type-

specific models were applied.

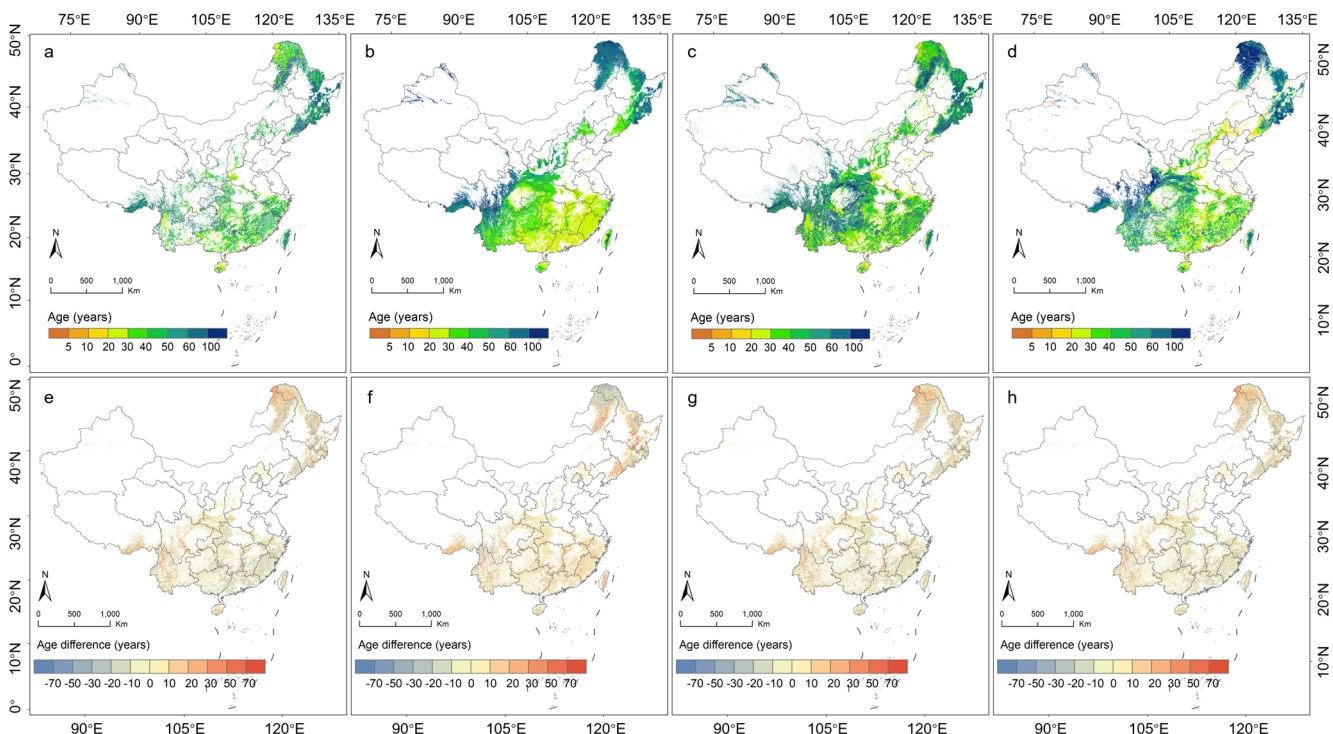

**Figure 10: Comparisons of the CAFA V2.0 forest age with previous forest age products. a**, **b**, **c**, and **d** are the age products in 2005, 2010, 2019, and 2020 from Zhang et al. (2014), Besnard et al. (2021), Shang et al. (2023), and Cheng et al. (2024), respectively. **e**, **f**, **g**, and **h** are the difference between the CAFA V2.0 forest age with previous four forest age products. The map lines may contain disputed territories.

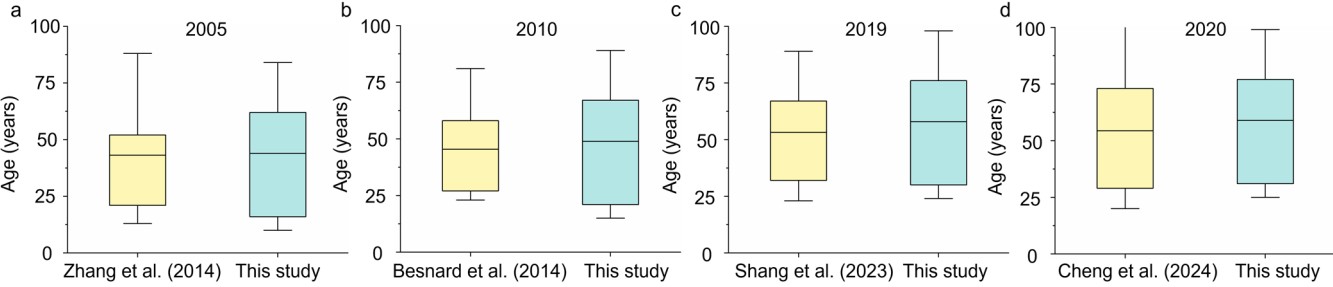

**Figure 11: Box plots of CAFA V2.0 forest age and four previous forest age products in China. a**, **b**, **c**, and **d** are Zhang et al. (2014), Besnard et al. (2021), Shang et al. (2023), and Cheng et al. (2024).

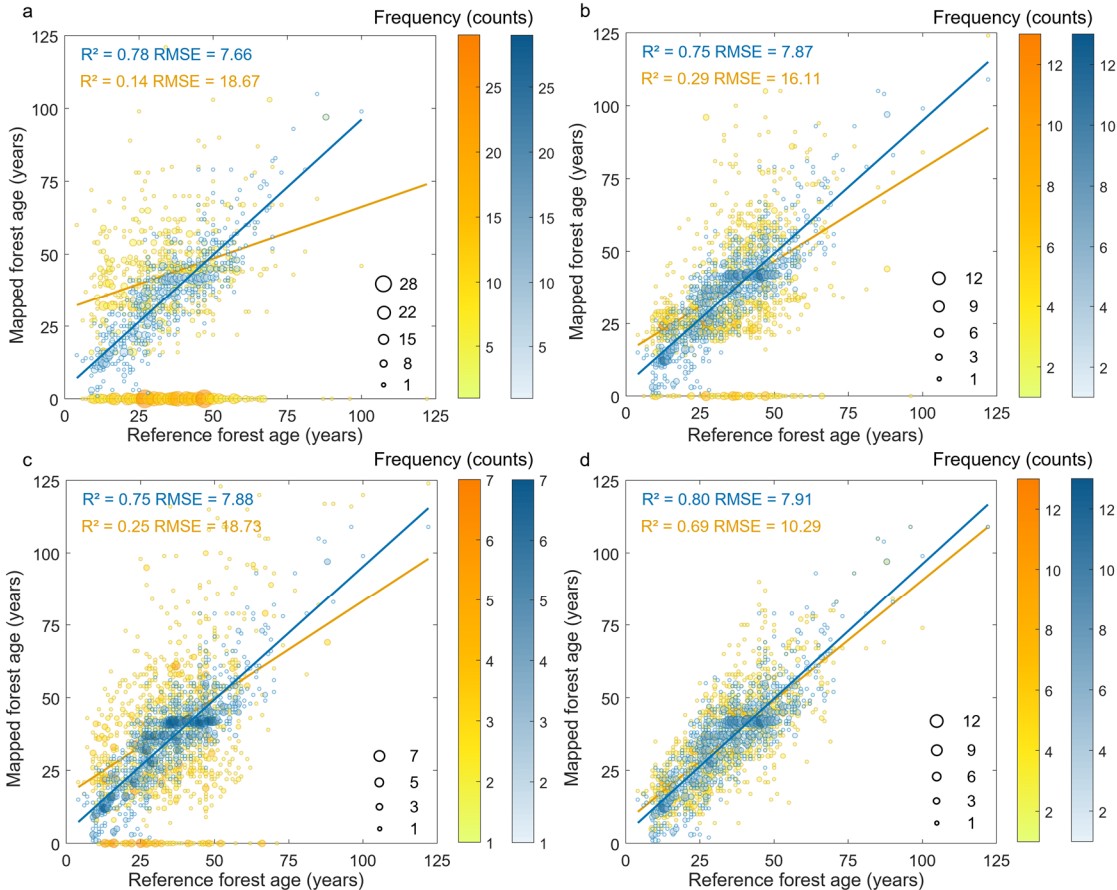

**Figure 12: Validation of the four forest age products using the same 30% reference forest age samples compared with the CAFA V2.0 product. a-d** are the forest ages of Age2005, Age2010, Age2020, and Age2019 generated by Zhang et al. (2014), Besnard et al. (2021), Cheng et al. (2024), and Shang et al. (2023), respectively. Blue represents the CAFA V2.0 product, while yellow represents the four products. The age of 0 in Age2005, Age2010, and Age2020 indicate no available forest age in these products for the validated reference samples, and they are excluded from the fitting line and the calculations of R² and RMSE. To maintain consistency, the validation of the CAFA V2.0 product also excluded these reference samples.

## 5 Discussions

### 5.1 Contributions of different input factors to forest age mapping

A total of 15 input factors were utilized to map the age of undisturbed forests across six regions and five forest cover types in China. To investigate the contribution of each factor to the model, we first calculated the absolute mean SHAP value and then calculated its ratio in all mean SHAP values for each forest cover type (Fig. 13). A larger ratio of absolute mean SHAP value indicates a greater impact of that factor on forest age mapping. The results revealed that forest height was the dominant factor and contributed more than 20% to forest age mapping for most forest types, aligning with previous findings (Lin et al., 2023;

Cheng et al., 2024). This was because forest height is a key indicator of forest structure, which reflects the maturity and biomass of the forest (Shugart et al., 2010). Two terrain factors, slope, and aspect, generally ranked second or third, suggesting that individual terrain factors had a larger impact on forest age mapping than individual climate factors. For NIRv, NDVI, and the eight climate factors, there were no clear pattern across the five forest cover types. Most of these factors rank in the middle to lower positions, and the differences in the absolute mean SHAP values among the climate factors were minimal. However,

there were some regional and forest cover type exceptions. For example, the lowest annual temperature contributed 21% to forest age mapping for deciduous broad-leaved forests in East China, likely because low temperatures limit the growth of certain tree species and affect vegetation composition and distribution (Gazol et al., 2022). In the case of evergreen coniferous forests in Northwest China, mean annual precipitation contributed 20.9% to forest age mapping. Precipitation is critical for supporting evergreen vegetation growth in this region, as it ensures both the productivity of evergreen forests and the stability

of the ecosystem (Duan et al., 2019). Soil type, on the other hand, did not have a significant impact, aligning with findings by Cheng et al. (2024), which suggest that soil type plays a relatively minor role in forest age mapping.

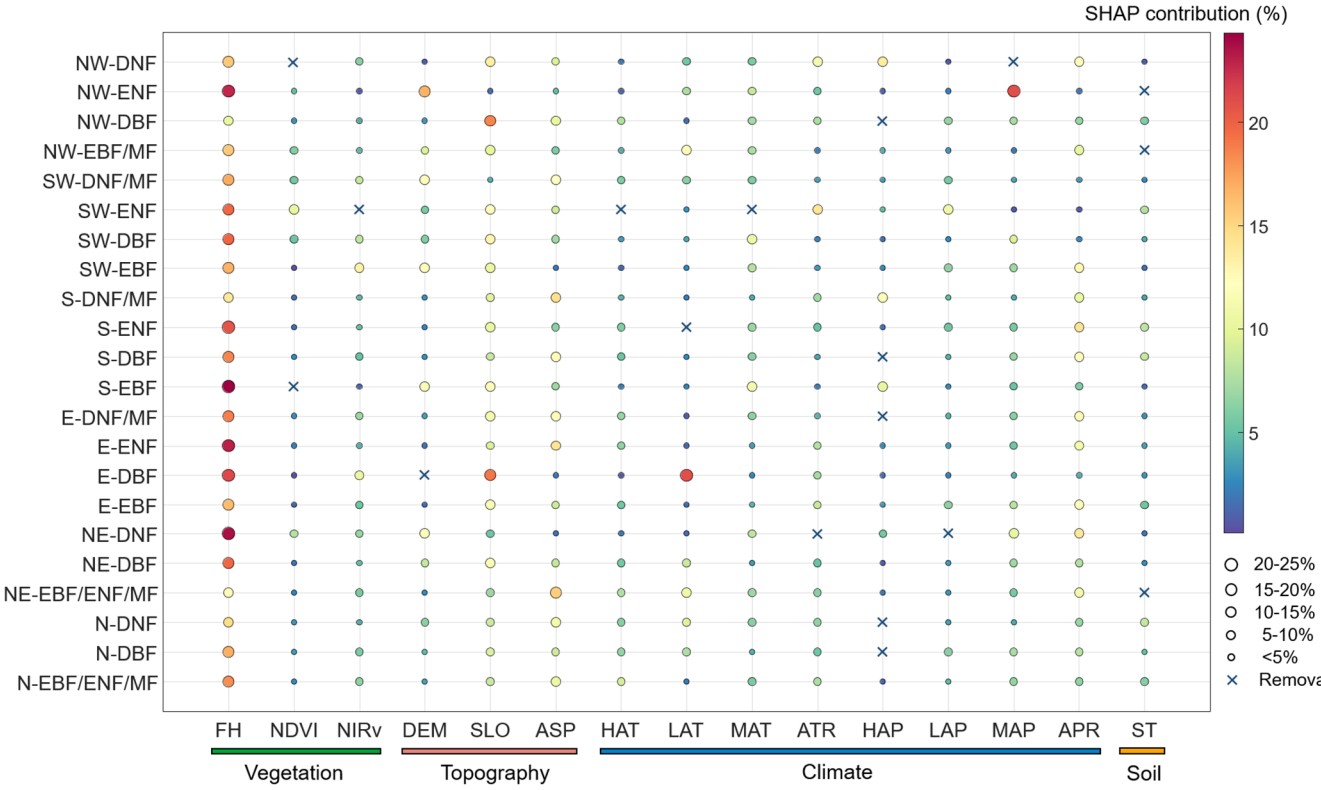

**Figure 13: Contributions of different input factors to forest age mapping for five forest cover types.** EBF: evergreen broad-leaved forest, DBF: deciduous broad-leaved forest, ENF: evergreen coniferous forest, DNF: deciduous coniferous forest, MF: mixed forest. N: Northern region, NE: Northeast region, E: East China, S: South China, SW: Southwest region, NW: Northwest region. FH is the forest height. NDVI is the normalized difference vegetation index. NIRv is the near-infrared reflectance of vegetation. DEM is the digital elevation model. SLO is the slope. ASP is the aspect. HAT is the highest annual temperature, LAT is the lowest annual temperature, MAT is the mean annual temperature, ATR is the annual temperature range, HAP is the highest annual precipitation, LAP is the lowest annual precipitation, MAP is the mean annual precipitation, APR is the annual precipitation range. ST is the soil type.

## 5.2 The impact of different forest height retrievals on age mapping

Forest height retrieval was necessary for age mapping under two situations: first, when forest height data were unavailable from the two forest height products in 2019, and second, when at least one disturbance occurred before 2019. Figure 14a presents the histogram of the percentage of pixels requiring forest height retrieval each year from 1986 to 2019. All years had percentages below 0.8%, with an average of 0.42%, indicating that only a small portion of the total pixels required height retrieval. It should be noted that the forest height retrieval used in this study differed from the 2019 forest height products, as the models were trained on different samples and input factors, potentially introducing uncertainties into forest age mapping. To assess the impact of varying forest height retrieval methods, 50% of the validating samples were randomly selected to compare forest age mapping based on the retrieved forest height in this study and the 2019 forest height product. Differences in forest height were observed between the two methods, which resulted in an RMSE of 5.92 years in forest age mapping (Fig.

14b). However, since 99.58% of pixels did not require forest height retrieval, we believe that the impact of these differences on forest age mapping in CAFA V2.0 is negligible.

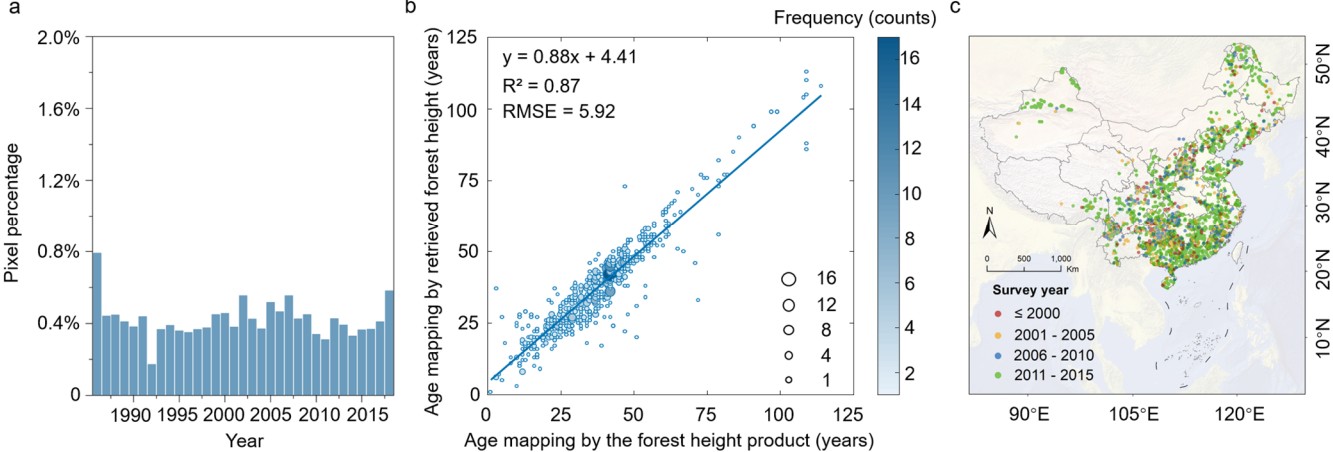

**Figure 14: The percentages of pixels needing forest height retrieval from 1986 to 2019 (a), comparisons of forest age mapping using the retrieved forest height versus forest height product in 2019 (b), and the spatial distribution of the survey years of samples used**
**for comparison (c).** The map lines may contain disputed territories.

### 5.3 Importance of forest age mapping for different forest cover types

The ages of undisturbed forests were mapped using separate RF models for different forest cover types in this study, as different forest cover types exhibited varying growth patterns and responses to environmental factors (Körner, 2007; Bazzaz, 1996). To demonstrate its applicability, we compared the forest age mapping with and without forest cover type classifications (Fig. 15),

using the same input factors, training samples, and validating samples. Results indicated that classifying forest cover types led to an increase of overall $R^2$ by 0.07 and a decrease of RMSE by 1.44. The accuracy for each forest type was also higher than that without forest cover type classification, with the accuracy for DNF increasing the most, as its RMSE decreased by 3.77. Therefore, classifying forest cover types was essential for improving the accuracy of national forest age mapping.

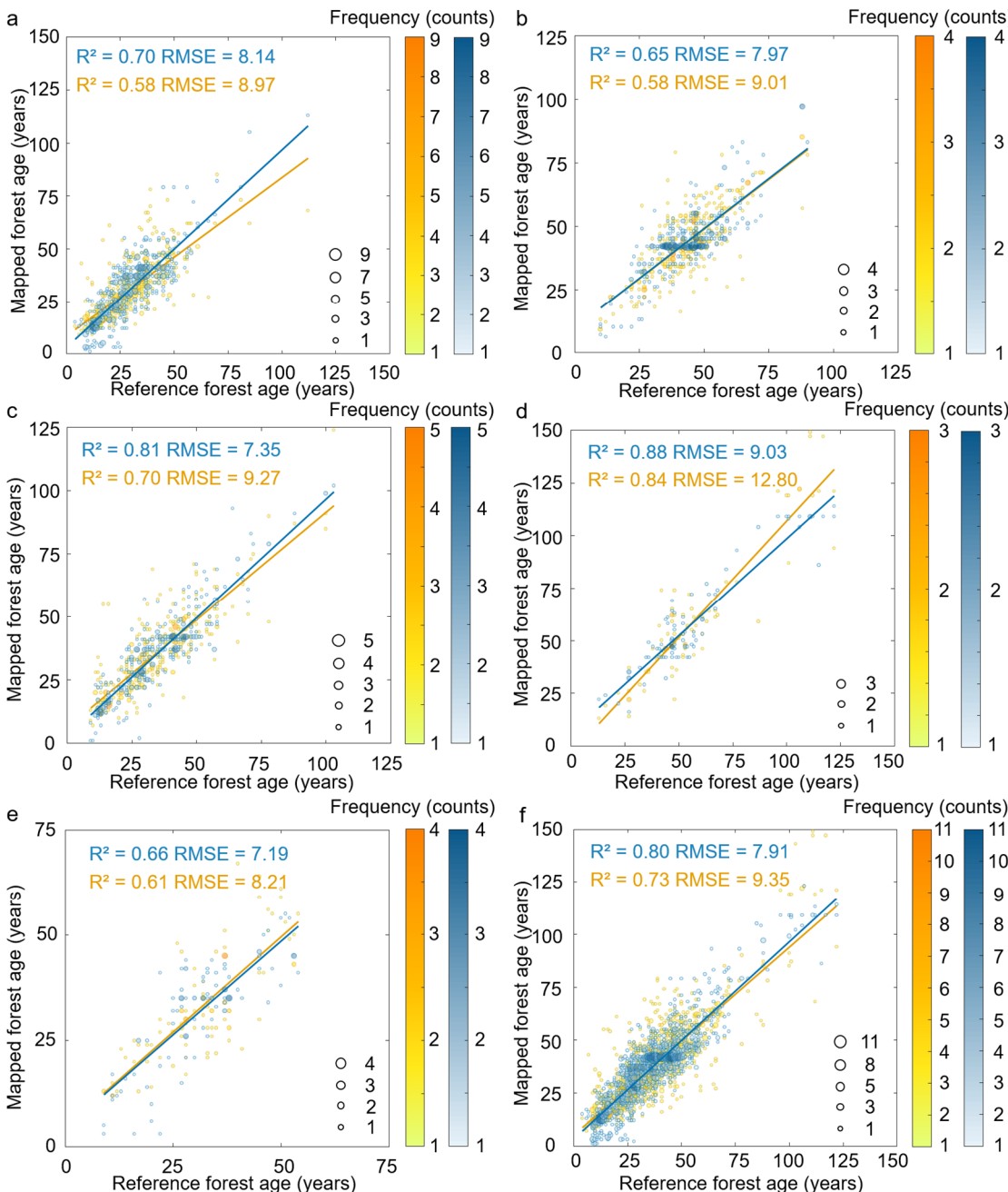

**Figure 15: Comparisons of the forest age mapping with and without forest cover type classifications. a** is evergreen broad-leaved forest, **b** is deciduous broad-leaved forest, **c** is evergreen coniferous forest, **d** is deciduous coniferous forest, **e** is mixed forest, **f** is all forest.

## 5.4 Limitations and future modifications

There were also some limitations in mapping forest age in CAFA V2.0. First, the year of disturbance may not always coincide with the year when forests are replanted or begin to recover. In this study, it was assumed that the forest age was 0 in the year

of disturbance, and 1 in the following year. However, delays in replanting or natural recovery after a disturbance could result in slight deviations in forest age estimates. The mCOLD forest disturbance monitoring algorithm can identify both the date of disturbance (t_break) and the date when vegetation recovery begins (t_start). Since t_break is determined with higher accuracy compared to t_start (Shang et al., 2025), it was used in this study for forest age mapping. To assess the potential impact of this assumption, the time interval between t_break and t_start was calculated for reference forest disturbance samples. Results

showed that 88.78% of reference samples had an interval of less than one year, 8% had an interval of 1-2 years, and 3.22% had an interval of more than two years. These findings suggest that using the disturbance year instead of the actual recovery year has only a minor impact on forest age mapping. Furthermore, planted young trees may already be older than one year, which could partially offset the discrepancies caused by the time difference between disturbance year and recovery year in forest age mapping.

Second, the number of reference forest age samples is limited. The reference forest age samples used in this study include 3,121 samples obtained from SPPCB field surveys (Fang et al., 2018) and an additional 2,183 samples obtained through the literature review (Luo et al., 2014; Cook-Patton et al., 2020). However, the samples, particularly those derived from the literature, were collected by various researchers using different methods to estimate forest age. This methodological variation may introduce some degree of bias into the final forest age mapping. Additionally, differences in sample distribution across

regions and forest cover types may also contribute to uncertainties in forest age estimation. Furthermore, due to the limited samples, especially in the northwest region, the ages of some very old trees may be underestimated. Most of these trees are found in high-altitude areas without competition from other vegetation (Liu et al., 2019), and their coverage is often smaller than the 30-meter resolution of Landsat pixels, resulting in their higher ages being averaged out. Therefore, future work should focus on collecting more field-based forest age reference samples to minimize the uncertainties in forest age mapping caused

by the reference sample limitations.

Third, the varied plot sizes of reference samples may influence forest age mapping, particularly for the 2,183 samples derived from literature reviews. While SPPCB samples had consistent plot sizes (primarily 1,000 m²), the plot sizes of the literature-derived samples varied. To assess these impacts, we analyzed spatial heterogeneity using a 100 m × 100 m window—much larger than the SPPCB plot size—with forest height standard deviation (SD) as the key metric. The results indicated that

89.1% of samples had low heterogeneity (SD < 2 m), corresponding to a mean forest age mapping difference of 5.7 years, which was smaller than the mapping error of 7.91 years for undisturbed forests. Moderate heterogeneity (SD 2–3 m) affected 7.33% of samples, causing a comparable difference of 8.3 years. High heterogeneity (SD > 3 m) was found in 3.67% of samples, leading to relatively larger differences of 13.3 years. Although 96.33% of samples had low to median heterogeneity with acceptable mapping errors, the identified high-heterogeneity samples (3.67%) caused relatively larger mapping errors of 13.3

570 years. Therefore, future studies should consider filtering out high-heterogeneity samples in forest age mapping.

Fourth, some of the input factors used in estimating the age of undisturbed forests with the boosted RF model may exhibit autocorrelation, potentially introducing biases into the forest age estimation. This study employed a total of 15 input factors for forest age mapping, including three vegetation factors, three topographic factors, eight climate factors, and one soil factor.

There may be some degree of autocorrelation within each category of input factors. To mitigate the effects of autocorrelation,
this study proposed an automatic iterative selection mechanism for input factors, excluding those contributing less than 0.5%
in each, and ultimately using the final model obtained after three iterations for forest age estimation. A comparison of the
accuracy of forest age estimations with and without the automatic iterative mechanism showed that the mechanism slightly
improved estimation accuracy, thereby reducing the impact of autocorrelation among input factors to some extent.

Fifth, other input data may also affect the forest age mapping. For instance, the original spatial resolution of the climate and
580 soil data was larger than 30 meters, and these disparities in spatial resolution were likely to introduce uncertainty. However,
due to the high spatial similarities of climate and soil within a small area, minimal variations are expected among nearby pixels.
Moreover, the contributions of these input factors to forest age mapping were relatively small (section 5.1). Therefore, their
impact on the accuracy of forest age mapping would not be significant.

Finally, due to data limitations, some input factors that could potentially enhance forest age estimation accuracy were not
included in this study, such as diameter at breast height (DBH), forest density, site index, and soil fertility (Chen et al., 2019;
Lin et al., 2023; Wylie et al., 2019). Our previous research in Fujian Province has shown that incorporating factors like DBH,
forest density, and site index can improve forest age estimation accuracy to some extent (Lin et al., 2023). However, since
these data are not available on a national scale, they were excluded from this study. As remote sensing technologies continue
to evolve, it is likely that these factors will eventually become estimable (Li et al., 2024; Socha et al., 2020; Fan et al., 2018),
further improving the accuracy of forest age mapping.

## 6 Data availability

The generated China's Annual Forest Age (CAFA V2.0) dataset at 30 m spatial resolution from 1986-2022 is publicly available
at https://doi.org/10.6084/m9.figshare.24464170 (Shang et al., 2023b).

## 7 Conclusions

This study generated China's annual forest age dataset (CAFA V2.0) at a 30-m resolution from 1986 to 2022, combining forest
disturbance monitoring and machine learning techniques. Forest disturbance monitoring, which has lower uncertainty than
machine learning, was used to update the annual forest age, employing the modified COLD (mCOLD) algorithm with
consideration of spatial variation within disturbance area and bidirectional time series tracking. For undisturbed forests, forest
age was estimated using machine learning models tailored to different regions and forest types, incorporating forest height,
vegetation indices, climate, terrain, and soil data. Adjustments were made for underestimations in the Northeast and Southwest
regions identified in CAFA V1.0, with additional reference samples and region-specific and forest type-specific models.
Validation showed that the mapped age of disturbed forests had a small error of ±2.48 years, while undisturbed forests had a

relatively large error of ±7.91 years. The generated 30 m annual forest age dataset can facilitate forest carbon cycle modelling in China, offering valuable insights for national forest management practices.

## Author contributions

Conceptualization, R.S. and J.C.; Methodology, X.L., R.S., J.C. and Y.L.; Validation, X.L. and R.S.; Formal analysis, X.L., K.F., M.X., Y.Y., R.L., Y.L., W.J., L.L., J.L., W.L. and J.Z.; Writing—original draft, R.S. and X.L.; Writing—review & editing, R.S., J.C. and M.X.; Funding acquisition, R.S., M.X. and Z.H.; Data curation, G. Y., N. H. and L. X..

## Code availability

The codes are available on request from the first and corresponding authors.

## Competing interests

The authors declare that they have no conflict of interest.

## Acknowledgments

We would like to express our sincere gratitude to all those who provided data and suggestions for this study.

## Financial supports

This research was funded by the National Natural Science Foundation of China (U23A2002, 42471356, 42101367, and 42201360), the Natural Science Foundation of Fujian Province (2021J05041), and the Open Fund Project of the Academy of Carbon Neutrality of Fujian Normal University (TZH2022-02).

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
