# Peer review of "China's annual forest age dataset at 30 m spatial resolution from 1986 to 2022"

_Earth System Science Data, 2024_

## Author Comment (AC1)

**Response to the reviewer's comments**

**Comments 1:**

How was the forest height estimated for years without GEDI footprints? Please explain clearly.

**Response:**

Thanks for your valuable suggestions. The forest height retrieval for years before 2019 at a specific forest type and region utilized the same model as 2019 but with varied inputs corresponding to their respective years. Related descriptions were revised in the manuscript.

*"To reduce the discrepancy between the retrieved forest height and the 2019 forest height products, the input factors for the tree height model were based on the study by Liu et al. (2022), incorporating DEM, slope, aspect, temperature, precipitation, and NDVI data. Additionally, this study expanded the input factors by incorporating Landsat 8 surface reflectance and the calculated NIRv vegetation index, which approximates forest productivity (Badgley et al., 2019). This resulted in a total of 13 input factors used in the tree height model. The models were constructed separately for different environmental conditions and forest types across China. Specifically, six regions (East, South, North, Northeast, Northwest, and Southwest) and five forest types (EBF, ENF, DBF, DNF, and MF) were considered. Each model was trained using 70% of the filtered GEDI footprint forest height samples, with the remaining 30% used for validation. Since no GEDI footprint forest height samples were available prior to 2019, the forest height retrieval for years before 2019 at a specific forest type and region utilized the same model as 2019 but with varied inputs corresponding to their respective years. These forests, which required height retrieval, covered only a small portion of the total forest pixels, with an average of 0.42% (see details in Section 5.2)."*

**Comments 2:**

Fig 6 and others:what does the circle size indicate?

**Response:**

Thanks for your valuable comments and suggestions. The size and color of the circle represent the number of samples at that location, and the related descriptions were revised in the manuscript.

*"**Figure 6: Validation of China's forest age mapping in the CAFA V2.0 dataset. a** is for the age validation of forests disturbed at least once from 1986 to 2022, and **b** is for the age validation of undisturbed forests. The size and color of the circle represent the number of samples at that location."*

**Comments 3:**

Fig 11: Add a validation figure for the result from this study as comparison.

**Response:**

Thanks for your valuable comments and suggestions, and it was added.

[Figure]

*"**Figure 12**: Validation of the four forest age products using the same 30% reference forest age samples **compared with the CAFA V2.0 product**. **a-d** are the forest ages of Age2005, Age2010, Age2020, and Age2019 generated by Zhang et al. (2014), Besnard et al. (2021), Cheng et al. (2024), and Shang et al. (2023), respectively. Blue represents the CAFA V2.0 product, while yellow represents the four products. The age of 0 in Age2005, Age2010, and Age2020 indicate no available forest age in these products for the validated reference samples, and they are excluded from the fitting line and the calculations of $R^2$ and RMSE. To maintain consistency, the validation of the CAFA V2.0 product also excluded these reference samples."*

**Comments 4:**

Fig 13:Please also add a figure show the distribution of forest height and age samples over different years.

**Response:**

Thanks for your valuable suggestions. A subfigure of the spatial distribution of the survey years of samples used for comparison was added.

[Figure]

*"Figure 14: The percentages of pixels needing forest height retrieval from 1986 to 2019 (a), comparisons of forest age mapping using the retrieved forest height versus forest height product in 2019 (b), and the spatial distribution of the survey years of samples used for comparison (c)."*

**Comments 5:**

Please also add a comparison between forest age map for planted forest and natural forest.

**Response:**

Thanks for your valuable comments and suggestions. It was added.

*"**4.1 China's annual forest age at 30m resolution from 1986 to 2022***

*A dynamic forest age dataset (CAFA V2.0) covering the entire China from 1986 to 2022 (Shang et al., 2023), with a spatial resolution of 30 m, was generated by integrating forest disturbance mapping and random forests methods. Figure 5 illustrates the distribution of forest ages for the year 2019, alongside comparisons with data from 1986, 2000, 2010, and 2022. This forest age dataset indicates that China's forest age structure predominantly consists of young and middle-aged forests, with an average forest age of 58.1 ± 7.3 years in 2019. Old forests were predominantly found in the northeast, northwest, and southwest regions of China. These areas, characterized by high mountains and minimal human interference, were largely comprised of natural and secondary forests. In contrast, forests disturbed at least once during the period from 1986 to 2022 exhibited younger ages, generally below 37 years. Such forests were mainly concentrated in the southeast and central southern regions, where human disturbances were more prevalent. Furthermore, in the northeast, there were also young forests that had regenerated after extensive forest fires, such as the devastating forest fire that occurred on May 6, 1987 (Cahoon Jr. et al., 1991). This fire caused varying degrees of damage across a vast area within China, impacting more than one million hectares of forests.*

*According to the 2020 map of China's planted and natural forests (Cheng et al., 2023), natural forests were older than planted forests, with average ages of 69.9 ± 12.8 years and 48.4 ± 6.9 years, respectively. Regional variations were evident, with the age gap between natural and planted forests ranging from 3.5 to 20.2 years. Southwest China had the oldest natural forests (91.1 ± 21.6 years) and planted forests (74.8 ± 18.1 years), while East and South China showed lower average ages due to higher disturbance frequencies."*

---

## Author Response (AR1)

**Summary of revisions:**

We have made five major revisions following the reviewers' suggestions, including: (1) The uncertainty analysis of the mapped forest age was added. (2) The age comparisons between China's planted and natural forests were added. (3) The anlysis of the temporal dynamics of China's forest ages was added. (4) The validations of the merged forest cover type dataset and orginal three forest cover products were added. (5) Many descriptions were revised and added to make the manuscript easier to understand. Moreover, we have followed the reviewers' suggestions and provided feedback to the reviewers on a point-by-point basis (see responses below).

**Response to the reviewer 1's comments**

**Comments 1.1:**

The introduction should further emphasize the significance of time-series forest age data, particularly in the context of forest ecosystem dynamics, carbon cycle modeling, and long-term forest management, to highlight the necessity and scientific value of this study.

**Response:**

Thanks for your valuable comments and suggestions. It was revised.

*"Several forest age products have been developed for China (Zhang et al., 2017, 2014; Xiao et al., 2023; Shang et al., 2023; Cheng et al., 2024; Besnard et al., 2021). Early studies produced three sets of forest age products with a spatial resolution of 1000 meters for the years 2005 and 2010 using the height-based method (Zhang et al., 2014; Zhang et al., 2017; Besnard et al., 2021). However, the 1000-meter resolution averages forest stands within each pixel, leading to overestimations of young forests and underestimations of old forests. In recent years, driven by the demand for precise simulation of forest carbon dynamics and the availability of high-resolution remote sensing data, several high-resolution forest age products have been successfully generated. For example, Xiao et al. (2023) estimated forest age in disturbance areas across China at a 30-meter resolution in 2020 using the CCDC disturbance monitoring algorithm. Cheng et al. (2024) combined machine learning algorithms based on tree height, climate, and terrain with the LandTrendr disturbance monitoring algorithm to obtain forest age data for China in 2020. Our previous work (Shang et al., 2023) utilized machine learning algorithms and the COLD disturbance monitoring algorithm to estimate nationwide forest age at a 30-m resolution in 2019 (CAFA V1.0). Compared to earlier products that used the height-based method alone, integrating it to estimate forest age with the disturbance-based method for updating forest age significantly enhances reliability. However, significant discrepancies still exist among current forest age products, which provide data for single years only, thus overlooking substantial changes in forest age before and after disturbances. These omitted changes can have a large impact on forest carbon modeling. When*

*using single-year forest age data, process-based ecosystem models often underestimate the forest carbon uptake prior to the most recent forest disturbance and fail to account for the carbon release from multiple forest disturbances, leading to substantial uncertainties in forest carbon modeling (Yu et al., 2020; Zhang et al., 2025). In contrast, long-term forest age products can capture these carbon dynamics, making them more valuable for forest carbon modeling and forest management (Chorshanbiyev et al., 2024; Zhang et al., 2025). Therefore, it is urgent to generate long-term, high-resolution forest age products to support China's carbon neutrality researches (Besnard et al., 2021; Schumacher et al., 2020; Yu et al., 2020)."*

**Comments 1.2:**

The source and spatiotemporal resolution of forest type data in Figure 1 need to be clarified. Please specify how this data was obtained and its respective temporal and spatial resolutions.

**Response:**

Thanks for your valuable comments and suggestions. Related descriptions were revised in the manuscript.

[revised manuscript text omitted]

**Comments 1.4:**

While the NDVI formula is well known, the calculation method for NIRv is not explicitly provided. It is recommended to include the formulas for both indices to facilitate a better understanding of their computation.

**Response:**

Thanks for your valuable comments and suggestions. The equations of NDVI and NIRv were added in section 3.1.3.

"**3.1.3 Mapping the age of undisturbed forests through random forests**

*Random forests with boosting trees (Jahan et al., 2021) were also selected to map the age of undisturbed forests. This machine learning method showed the highest overall accuracy in forest age mapping at a 30 m spatial resolution among five stand growth equations and twelve machine learning methods (Lin et al., 2023). The model used fifteen inputs (Table 1): vegetation factors (forest height, NDVI, and NIRv), terrain factors (slope and aspect), climate factors (HAT, LAT, MAT, ATR, HAP, LAP, and MAP), and one soil factor (soil type). Tree height was selected due to its dominant role in forest age mapping (Lin et al., 2023), while NDVI and NIRv (Equations (1) and (2)) could reflect forest greenness and productivity. Terrain factors such as slope and aspect also affected forest growth (Lang et al., 2010). Temperature and precipitation were included as forest growth is sensitive to climatic conditions (Besnard et al., 2021). Soil type was also considered for its effect on vegetation growth.*

$$NDVI = \frac{NIR_{Ref} - RED_{Ref}}{NIR_{Ref} + RED_{Ref}} \tag{1}$$

$$NIR_V = NIR_{Ref} * NDVI \tag{2}$$

*where, $NIR_{Ref}$ represents the surface reflectance at the near-infrared band, and $RED_{Ref}$ represents the surface reflectance at the red band."*

**Comments 1.5:**

What are the plot sizes for the two types of sample datasets? Are they consistent? If not, could this discrepancy impact the accuracy or consistency of the forest age estimation? A discussion on this issue would be beneficial.

**Response:**

Thanks for your valuable suggestions. The SPPCB samples had consistent plot sizes (primarily 1,000 m²), while the plot sizes of the literature-derived samples varied. 96.33% of samples had low

to median heterogeneity with acceptable age mapping errors, but the identified high-heterogeneity samples (3.67%) caused relatively larger mapping errors of 13.3 years. Related descriptions were also added to the discussion.

*"**5.4 Limitations and future modifications***

*……*

*Third, the varied plot sizes of reference samples may influence forest age mapping, particularly for the 2,183 samples derived from literature reviews. While SPPCB samples had consistent plot sizes (primarily 1,000 m²), the plot sizes of the literature-derived samples varied. To assess these impacts, we analyzed spatial heterogeneity using a 100 m × 100 m window—much larger than the SPPCB plot size—with forest height standard deviation (SD) as the key metric. The results indicated that 89.1% of samples had low heterogeneity (SD < 2 m), corresponding to a mean forest age mapping difference of 5.7 years, which was smaller than the mapping error of 7.91 years for undisturbed forests. Moderate heterogeneity (SD 2–3 m) affected 7.33% of samples, causing a comparable difference of 8.3 years. High heterogeneity (SD > 3 m) was found in 3.67% of samples, leading to relatively larger differences of 13.3 years. Although 96.33% of samples had low to median heterogeneity with acceptable mapping errors, the identified high-heterogeneity samples (3.67%) caused relatively larger mapping errors of 13.3 years. Therefore, future studies should consider filtering out high-heterogeneity samples in forest age mapping."*

**Comments 1.6:**

How was the quality of the forest disturbance samples ensured? Was independent validation performed? Providing relevant validation methods would enhance the credibility of the sample data.

**Response:**

Thanks for your valuable comments and suggestions. The quality of the forest disturbance samples was ensured through a multi-stage interpretation process involving multiple experts. In the first stage, samples were divided into sets of 1,000, with each set being independently interpreted by at least three of the 13 experts. This initial round identified samples with unanimous agreement among the experts, which were accepted as final. In the second stage, samples with partial agreement were reviewed by additional experts to reach a consensus. Finally, any remaining samples were voted on by all experts, with those receiving over 50% of the votes being accepted as final.

The independent validation was mainly conducted in the first stage of the forest disturbance interpretation, showing a consistency rate ranging from 43% to 81%. Related descriptions were revised in the manuscript.

*"The reference forest disturbance samples (Fig. 2b) were used to validate the ages of forests disturbed at least once between 1986 and 2022. The age of these samples was derived from the*

*number of years since the disturbance event. The reference forest disturbance samples were interpreted through analysis of time series images from Google Earth, PlanetScope, Sentinel-2, or Landsat 5/7/8, with each event confirmed by at least two clear-sky images taken before and after the disturbance (Qiu et al., 2023; Shang et al., 2025). The interpretation of reference forest disturbance samples was performed in three stages. Initially, samples were divided into sets of 1,000, with each of the 13 experts independently interpreting three sets. This ensured that each set was reviewed by at least three experts. For each set, samples unanimously identified by three experts (consistency rate 43%–81%) were accepted as final. In the second stage, samples identified by two experts were reviewed by a fourth expert, while those with no consensus were reviewed by both a fourth and fifth expert. Samples confirmed by at least three experts were accepted as final. In the final stage, the remaining unconfirmed samples were voted on by all experts, with those receiving over 50% of the votes being accepted as final. A total of 12,328 forest disturbance samples were interpreted, with 4,168 samples having at least one forest disturbance. Of these, 2,157 samples experienced a single disturbance event between 1986 and 2022, 1,274 points had two disturbances, and 737 points were disturbed more than twice. "*

**Comments 1.7:**

The Liu and Potapov forest height products have differences in forest extent and definitions. How did the authors address this inconsistency? Additionally, it was mentioned that 0.32% of pixels lacked forest height values—what dataset was used as the baseline for defining forest areas in this case?

**Response:**

Thanks for your valuable comments. Liu's and Potapov's forest height products use the same definition of forests, but they have different forest extents. Liu's product has a smaller forest extent compared to Potapov's product because it uses a more rigorous quality control standard. Therefore, this study mainly used Liu's forest height product. When Liu's product was missing compared with the forest extent identified in the China Land Cover Dataset (CLCD) (Yang and Huang, 2021), Potapov's forest height product was used as a substitute. If both Liu's and Potapov's products were missing, a forest height inversion model was developed to estimate the forest height.

The 0.32% of pixels with missing forest heights were identified using the CLCD dataset (Yang and Huang, 2021). Related descriptions in the manuscript were also revised.

*"**2.2.4 Forest height data***

*Two forest height products with the same forest definition at a 30-meter spatial resolution for the year 2019 (Potapov et al., 2021; Liu et al., 2022) were employed to map the age of undisturbed forests. Potapov et al. (2021) utilized machine learning methods with Landsat data and Global Ecosystem Dynamics Investigation (GEDI) footprint forest height data to generate a global forest*

*canopy height map at a 30-meter spatial resolution for the year 2019 (shortened to Potapov's forest height product), while Liu et al. (2022) developed a neural network guided interpolation (NNGI) method to derive China's forest height map at 30-meter spatial resolution for 2019 (shortened to Liu's forest height product), using Landsat data along with GEDI and ICESat-2 footprint forest height data. Due to consideration of topographic influences and high-quality control standards, Liu's forest height product exhibited higher accuracy but had a smaller forest extent in China than Potapov's forest height product (Liu et al., 2022). Therefore, this study primarily used Liu's forest height product. When Liu's product was missing compared with the forest extent identified in CLCD, Potapov's forest height product was used as a substitute.*

*For forest pixels with missing forest heights from both Liu's and Potapov's two products (0.32% of pixels, based on the forest extent inCLCD), forest height was estimated using a machine learning method (detailed in Section 3.2.2) that integrates Landsat data, climate data, terrain data, and GEDI footprint forest height data. The input Landsat data consists of surface reflectance from Landsat 5, 7, and 8 and two calculated vegetation indices (NDVI and NIRv). The input data from GEDI, launched by NASA in December 2018 and covering the Earth's land surface from 51.6°N to 51.6°S (Dubayah et al., 2020), primarily includes the L2A relative height data, which has demonstrated the best performance in global forest height mapping (Potapov et al., 2021)."*

**Comments 1.8:**

Has the synthesized forest type distribution dataset undergone independent validation? Furthermore, is the definition and extent of forests in this dataset consistent with the CLCD dataset used in the study? Further clarification is needed.

**Response:**

Thanks for your valuable comments and suggestions. We added validations of the merged dataset and the three forest cover type datasets using the field survey data from the SPPCB project (**Response Fig. 1**). GLC_FCS30 had the highest accuracy indicated by the Kappa coefficient, and ESA CCI LC had the lowest accuracy among the three forest cover type dataset. Compared with the three original datasets, the accuracy of our merged forest cover type dataset improved significantly, with a 3.2% higher Kappa coefficient than GLC_FCS30, 6.31% higher than GLASS-GLC, and 8.4% higher than ESA CCI LC.

The merged forest cover type dataset does differ from the CLCD dataset. CLCD is superior to the merged forest cover type dataset in the division of forest extent, while the merged forest cover type product was mainly used as inputs for forest age mapping.

Related descriptions in the manuscript were also revised.

[Figure]

**Response Fig. 1.** Confusion matrix of the merged dataset and the three forest cover type datasets. a: GLC_FCS30, b: GLASS-GLC, c: ESA CCI LC, and d: the merged dataset. The colors indicate the number of sample points.

*"**2.2.3 Forest extent and** forest cover type data*

*The China Land Cover Dataset (CLCD) (Yang and Huang, 2021) was used to indicate the dynamic forest extent of the forest age product. This dataset provides annual land cover information including forest cover extent for China from 1985 to 2022 at a 30 m spatial resolution, generated using Landsat imagery and random forest classifiers. It also had a comparable reliability to Hansen's Global Forest Change (GFC) dataset (Hansen et al., 2013) in terms of indicating forest changes (Yang and Huang, 2021). Several studies have also demonstrated that the CLCD offers higher accuracy than other land cover products across China (Zhang et al., 2022; Ji et al., 2024).*

*A merged forest cover type dataset (Fig. 1) was used for forest age mapping, as forest age mapping requires forest cover types as inputs, which were not provided by the CLCD product. This dataset was merged from three forest cover type products (Shang et al., 2023): GLC_FCS30 from 1985 to 2022 (Zhang et al., 2021c) at the 30 m resolution, GLASS-GLC from 1985 to 2020 (Liu et al., 2021) at the 30 m resolution, and ESA CCI LC from 1992 to 2019 (ESA, 2017) resampled into the 30 m resolution from the 300 m resolution. There were four merging rules: first, a forest type was designated if at least two products identified the same forest cover type; second, if all three*

*products had different types, the type from GLC_FCS30 was used, as it closely matched China's ninth forest resource report; third, if GLC_FCS30 indicated non-forest, the type from GLASS-GLC was used due to its higher spatial resolution than ESA CCI; last, if both GLC_FCS30 and GLASS-GLC indicated non-forest, the type from ESA CCI was utilized.* *The merged dataset and the three forest cover type datasets were validated against the field forest cover type data from the SPPCB project (Fang et al., 2018), and the accuracy of the merged dataset improved significantly. Specifically, the Kappa coefficient of the merged dataset was 3.2% higher than that of GLC_FCS30, 6.31% higher than GLASS-GLC, and 8.4% higher than ESA CCI LC."*

**Comments 1.9:**

Did the authors obtain a time-series forest height dataset for undisturbed areas?

**Response:**

Thanks for your valuable comments. No, we didn't. For the undisturbed forests, only the forest height for the year 2019 was required to estimate the forest age in 2019 ("baseline" age), and the time series ages for these undisturbed forests were then updated based on the number of years since 2019 (Fig. 4).

**Comments 1.10:**

Are the parameters of the random forest algorithm consistent across models? Were hyperparameter optimizations conducted? Detailed information on parameter selection and tuning methods should be provided. Additionally, what tools were used for data processing? More technical details would improve reproducibility.

**Response:**

Thanks for your valuable comments and suggestions. To clarify, different models indeed have distinct parameters. As illustrated in **Table 2**, we also performed sensitivity analysis to determine the optimal thresholds for the minimum leaf size and number of trees for each model. The minimum leaf size was varied from 1 to 30, with an interval of 5, while the number of trees was adjusted from 50 to 300, with an interval of 50. The optimal thresholds were identified as those corresponding to the minimum RMSE of the mapped forest age.

MATLAB was used as the tool for building the forest age mapping models and mapping the annual forest ages. Related descriptions in the manuscript were also revised, and **Table 2** was also added to the manuscript.

*Table 2 : Parameters of the forest age mapping models for different regions and forest cover types.*
*EBF: evergreen broad-leaved forest, DBF: deciduous broad-leaved forest, ENF: evergreen*

*coniferous forest, DNF: deciduous coniferous forest, MF: mixed forest. N: Northern region, NE: Northeast region, E: East China, S: South China, SW: Southwest region, NW: Northwest region.*

| Models (region and forest type) | Minimum leaf size | Number of trees | Number of features |
|---|---|---|---|
| NW-DNF | 5 | 150 | 13 |
| NW-ENF | 5 | 150 | 14 |
| NW-DBF | 5 | 150 | 14 |
| NW-EBF/MF | 5 | 100 | 14 |
| SW-DNF/MF | 5 | 200 | 15 |
| SW-ENF | 5 | 100 | 12 |
| SW-DBF | 10 | 100 | 15 |
| SW-EBF | 10 | 150 | 15 |
| S-DNF/MF | 20 | 100 | 15 |
| S-ENF | 5 | 100 | 14 |
| S-DBF | 5 | 150 | 14 |
| S-EBF | 10 | 100 | 14 |
| E-DNF/MF | 20 | 50 | 14 |
| E-ENF | 10 | 100 | 15 |
| E-DBF | 10 | 100 | 14 |
| E-EBF | 10 | 100 | 15 |
| NE-DNF | 20 | 100 | 13 |
| NE-DBF | 20 | 100 | 15 |
| NE-EBF/ENF/MF | 5 | 200 | 14 |
| N-DNF | 10 | 200 | 14 |
| N-DBF | 5 | 150 | 14 |
| N-EBF/ENF/MF | 5 | 150 | 15 |

"*Given the regional and forest cover type heterogeneity across China, this study divided the model construction into six regions and five forest cover types. An RF model was developed using MATLAB for each forest cover type within each region. In regions where certain forest cover types had fewer than 30 sample points, those regions were merged, resulting in a total of 22 forest age models. This stratified approach aims to enhance the accuracy of forest age mapping as forest types can also influence the accuracy of forest age mapping (Lin et al., 2023). For each forest cover type within a region, 70% of the randomly selected reference forest age samples were used for model training, while the remaining 30% were used for validation. To mitigate the effects of autocorrelation between the input factors, this study proposed an automatic iterative selection mechanism, aimed at reducing autocorrelation by minimizing the number of input factors. The process began by using all input factors to estimate forest age, followed by extracting the*

*contribution weights of each factor. Factors with a contribution weight of less than 0.5% were removed. The remaining factors were then used to re-estimate forest age. This iterative process was repeated three times, with factors contributing less than 0.5% being excluded in each iteration, and the final model after three iterations was used for forest age estimation. In addition to input feature screening, we also performed sensitivity analysis to determine the optimal thresholds for the minimum leaf size and number of trees for each model. The minimum leaf size was varied from 1 to 30, with an interval of 5, while the number of trees was adjusted from 50 to 300, with an interval of 50. The optimal thresholds were identified as those corresponding to the minimum RMSE of the mapped forest age (Table 2). "*

**Comments 1.11:**

How was uncertainty evaluated? The paper does not mention relevant methods. It is recommended to include a quantitative analysis of uncertainty in forest age estimation.

**Response:**

Thanks for your valuable comments and suggestions. The uncertainty analysis was added.

*"**3.3 Uncertainty analysis***

*The uncertainty analysis primarily focused on the mapped ages of undisturbed forests in 2019 using the height-based method, as the disturbance-based method generally has lower uncertainties in mapping forest ages compared to the height-based method (Shang et al., 2023). The uncertainties of the mapped forest ages mainly stem from the models and their inputs. Among the model inputs, forest height has the most significant impact on forest age mapping (see section 5.1 for details). Therefore, we concentrated solely on the uncertainty of input forest height in forest age mapping. The evaluation of the differences between the estimated and product-based forest heights and their impact on forest age mapping is discussed in section 5.2. To assess the uncertainties of the age mapping models, we kept the inputs constant while varying the forest heights estimated by forest stand growth equations. Based on Zhang's forest stand growth equations in China (Zhang et al., 2014), we calculated the relative forest heights for the years 2017, 2018, 2020, and 2021, according to the region and forest types. These forest heights were then used as inputs to map forest ages, and their standard deviation was calculated to represent the uncertainties.*"

*"**4.1 China's annual forest age at 30m resolution from 1986 to 2022***

*......*

*The uncertainty analysis was performed on the mapped ages of undisturbed forests in 2019 (Fig. 6). In most regions, the mapped forest age in 2019 exhibited relatively low uncertainty, with an average uncertainty of 8.7 years across China. However, the southwest region displayed higher uncertainty, exceeding 40 years in Tibet and certain mountainous areas of Sichuan province. This*

*elevated uncertainty may be attributed to the heightened sensitivity of age mapping models to forest height in the southwest region (see section 5.1 for details). Additionally, the significant increase in forest height with age, as described by the forest stand growth equations in these areas (Zhang et al., 2014), further contributed to the increased uncertainty. Despite these regional variations, the mapped forest age in 2019 was generally stable and characterized by small uncertainties.”*

[Figure]

*Figure 6: Spatial distributions of the uncertainty of mapped forest age in 2019. The map lines do not necessarily depict accepted national boundaries.*

*“**5.4 Limitations and future modifications***

*……*

*Fifth, other input data may also affect the forest age mapping. For instance, the original spatial resolution of the climate and soil data was larger than 30 meters, and these disparities in spatial resolution were likely to introduce uncertainty. However, due to the high spatial similarities of climate and soil within a small area, minimal variations are expected among nearby pixels. Moreover, the contributions of these input factors to forest age mapping were relatively small (section 5.1). Therefore, their impact on the accuracy of forest age mapping would not be significant.”*

**Comments 1.12:**

It is suggested to increase the number of panels in Figure 5 to present more time-series forest age data and provide a more detailed analysis of its temporal dynamics to enhance the scientific

value of the dataset.

**Response:**

Thanks for your valuable comments and suggestions. The forest age maps for 2000 and 2010 and the temporal dynamics of forest ages were also added to the manuscript.

[Figure]

*Figure 5: Spatial distribution of China's forest age in 2019 and other selected years (1986, 2000, 2010, and 2022) in the CAFA V2.0 dataset. The map lines do not necessarily depict accepted national boundaries.*

*"**4.1 China's annual forest age at 30m resolution from 1986 to 2022**

A dynamic forest age dataset (CAFA V2.0) covering the entire China from 1986 to 2022 (Shang et al., 2023), with a spatial resolution of 30 m, was generated by integrating forest disturbance mapping and random forests methods. Figure 5 illustrates the distribution of forest ages for the year 2019, alongside comparisons with data from 1986, 2000, 2010, and 2022. This forest age dataset indicates that China's forest age structure predominantly consists of young and middle-aged forests, with an average forest age of 58.1 ± 7.3 years in 2019. Old forests were predominantly found in the northeast, northwest, and southwest regions of China. These areas, characterized by high mountains and minimal human interference, were largely comprised of natural and secondary forests. In contrast, forests disturbed at least once during the period from 1986 to 2022 exhibited younger ages, generally below 37 years. Such forests were mainly concentrated in the southeast and central southern regions, where human disturbances were more prevalent. Furthermore, in the northeast, there were also young forests that had regenerated after extensive forest fires, such as the devastating forest fire that occurred on May 6, 1987 (Cahoon Jr. et al., 1991). This fire caused varying degrees of damage across a vast area within China, impacting more than one million hectares of forests.*

*According to the 2020 map of China's planted and natural forests (Cheng et al., 2023), natural forests were older than planted forests, with average ages of 69.9 ± 12.8 and 48.4 ± 6.9 years,*

*respectively. Regional variations were evident, with the age gap between natural and planted forests ranging from 3.5 to 20.2 years. Southwest China had the oldest natural forests (91.1 ± 21.6 years) and planted forests (74.8 ± 18.1 years), while East and South China showed lower average ages due to higher disturbance frequencies.*

*The temporal dynamics of China's forest ages were primarily influenced by both forest loss disturbances (such as forest fire, harvest, and other disturbances) and forest gain disturbances (such as afforestation and reforestation), which mainly led to a reduction in China's average forest age. From 1987 to 2022, the age reduction caused by forest disturbances showed a decreasing trend, with an average age reduction of -0.105 ± 0.027 years. However, in 1987, 2008, and 2021, the forest age reduction caused by disturbances was significant, indicating that there were more forest disturbances in these three years."*

**Comments 1.13:**

Which year does the validation result in Figure 6 correspond to? Please specify this to ensure a clear understanding of the temporal relevance of the validation.

**Response:**

Thanks for your valuable suggestions. The validation year is 2019, and the related descriptions were revised in the manuscript.

*"**4.2 Validation of the forest age maps***

*The mapped forest age in 2019 was validated using 30% of two separate reference datasets (Fig. 6): one comprising 12,328 interpreted reference forest disturbance datasets and the other consisting of 5,304 forest field survey samples in China. For undisturbed forests, the field-surveyed age was transformed from the survey year to 2019 by adding the difference in years. For disturbed forests, the reference age in 2019 was determined by calculating the number of years since the last disturbance. Validation results indicated that the mapped age of disturbed forest exhibited a small error of ±2.48 years, while the mapped age of undisturbed forest from 1986 to 2022 had a relatively large error of ±7.91 years. Compared to version 1.0, the RMSE of CAFA V2.0 forest age for disturbed forests decreased by 1.15, and for undisturbed forests, the RMSE decreased by 0.49. "*

**Comments 1.14:**

The paper lacks a quantitative assessment and discussion of data uncertainty. It is suggested to incorporate uncertainty evaluation in the results or discussion sections to improve the study's completeness.

**Response:**

Thanks for your valuable comments and suggestions. The uncertainty analysis was added.

*"**3.3 Uncertainty analysis***

[revised manuscript text omitted]

**Comments 2.4:**

Fig 13:Please also add a figure show the distribution of forest height and age samples over different years.

**Response:**

Thanks for your valuable suggestions. A subfigure of the spatial distribution of the survey years of samples used for comparison was added.

[Figure]

*"Figure 14: The percentages of pixels needing forest height retrieval from 1986 to 2019 (a), comparisons of forest age mapping using the retrieved forest height versus forest height product in 2019 (b), and the spatial distribution of the survey years of samples used for comparison (c)."*

**Comments 2.5:**

Please also add a comparison between forest age map for planted forest and natural forest.

**Response:**

Thanks for your valuable comments and suggestions. It was added.

*"**4.1 China's annual forest age at 30m resolution from 1986 to 2022***

*A dynamic forest age dataset (CAFA V2.0) covering the entire China from 1986 to 2022 (Shang et al., 2023), with a spatial resolution of 30 m, was generated by integrating forest disturbance mapping and random forests methods. Figure 5 illustrates the distribution of forest ages for the year 2019, alongside comparisons with data from 1986, 2000, 2010, and 2022. This forest age dataset indicates that China's forest age structure predominantly consists of young and middle-aged forests, with an average forest age of 58.1 ± 7.3 years in 2019. Old forests were predominantly found in the northeast, northwest, and southwest regions of China. These areas, characterized by high mountains and minimal human interference, were largely comprised of natural and secondary forests. In contrast, forests disturbed at least once during the period from 1986 to 2022 exhibited younger ages, generally below 37 years. Such forests were mainly concentrated in the southeast and central southern regions, where human disturbances were more prevalent. Furthermore, in the northeast, there were also young forests that had regenerated after extensive forest fires, such as the devastating forest fire that occurred on May 6, 1987 (Cahoon Jr. et al., 1991). This fire caused varying degrees of damage across a vast area within China, impacting more than one million hectares of forests.*

*According to the 2020 map of China's planted and natural forests (Cheng et al., 2023), natural forests were older than planted forests, with average ages of 69.9 ± 12.8 years and 48.4 ± 6.9 years,*

*respectively. Regional variations were evident, with the age gap between natural and planted forests ranging from 3.5 to 20.2 years. Southwest China had the oldest natural forests (91.1 ± 21.6 years) and planted forests (74.8 ± 18.1 years), while East and South China showed lower average ages due to higher disturbance frequencies."*